# Plasma growth hormone pulses induce male-biased pulsatile chromatin opening and epigenetic regulation in adult mouse liver

Andy Rampersaud, Jeannette Connerney, David J Waxman*

Department of Biology and Bioinformatics Program, Boston University, Boston, United States

**\*For correspondence:**
djw@bu.edu

**Competing interest:** The authors declare that no competing interests exist.

**Abstract** Sex differences in plasma growth hormone (GH) profiles, pulsatile in males and persistent in females, regulate sex differences in hepatic STAT5 activation linked to sex differences in gene expression and liver disease susceptibility, but little is understood about the fundamental underlying, GH pattern-dependent regulatory mechanisms. Here, DNase-I hypersensitivity site (DHS) analysis of liver chromatin accessibility in a cohort of 18 individual male mice established that the endogenous male rhythm of plasma GH pulse-stimulated liver STAT5 activation induces dynamic, repeated cycles of chromatin opening and closing at several thousand liver DHS and comprises a novel mechanism conferring male bias to liver chromatin accessibility. Strikingly, a single physiological replacement dose of GH given to hypophysectomized male mice restored, within 30 min, liver STAT5 activity and chromatin accessibility at 83% of the dynamic, pituitary hormone-dependent male-biased DHS. Sex-dependent transcription factor binding patterns and chromatin state analysis identified key genomic and epigenetic features distinguishing this dynamic, STAT5-driven mechanism of male-biased chromatin opening from a second GH-dependent mechanism operative at static male-biased DHS, which are constitutively open in male liver. Dynamic but not static male-biased DHS adopt a bivalent-like epigenetic state in female liver, as do static female-biased DHS in male liver, albeit using distinct repressive histone marks in each sex, namely, H3K9me3 at male-biased DHS in female liver and H3K27me3 at female-biased DHS in male liver. Moreover, sex-biased H3K36me3 marks are uniquely enriched at static sex-biased DHS, which may serve to keep these sex-dependent hepatocyte enhancers free of H3K27me3 repressive marks and thus constitutively open. Pulsatile chromatin opening stimulated by endogenous, physiological hormone pulses is thus one of two distinct GH-determined mechanisms for establishing widespread sex differences in hepatic chromatin accessibility and epigenetic regulation, both closely linked to sex-biased gene transcription and the sexual dimorphism of liver function.

## eLife assessment

This **important** study offers new and **convincing** support for the idea that about a third of mouse liver DNAse-I hypersensitivity sites (DHS) showing male-biased chromatin opening are sex-biased because of the male-specific cyclic action of growth hormone pulses to alter chromatin accessibility compared to the relative ineffectiveness of the more static pattern of growth hormone secretion in females. Supporting evidence is found in the impact of hypophysectomy and growth hormone treatment on chromatin accessibility, and the binding of specific transcription factors and epigenetic marks at STAT5-sensitive sites. This work uncovers mechanisms underlying sex differences in liver function and will be of broad interest to endocrinologists and hepatologists.

## Introduction

Growth hormone (GH) regulates hepatic expression of enzymes and transporters that play critical roles in lipid metabolism (*Vázquez-Borrego et al., 2021*) and in the detoxification of many drugs and other lipophilic foreign chemicals (*Waxman and Holloway, 2009*). Dysregulation of hepatic GH signaling can lead to liver metabolic disorders, including the development of fatty liver disease and non-alcoholic steatohepatitis, with males more susceptible than females, as seen in both mice and humans (*Dichtel et al., 2022*; *Kaltenecker et al., 2019*; *Oxley et al., 2023*). Correspondingly, GH regulates many liver-expressed genes in a sex-dependent manner, enabling each sex to meet its specific metabolic and hormonal requirements (*Wauthier et al., 2010*). This program of sex-dependent gene expression is controlled by the sex-dependent temporal patterns of pituitary GH secretion (*Brie et al., 2019*; *Farhy et al., 2007*), which emerge at puberty but are programmed earlier in life by neonatal exposure to androgen (*Chowen et al., 1996*; *Ramirez et al., 2010*; *Waxman et al., 1985*). Pituitary GH secretion and, consequently, plasma GH profiles are intermittent (pulsatile) in pubertal and adult males, whereas they are near-continuous (persistent) in pubertal and adult females, as seen in rats, mice, and humans (*Waxman and O'Connor, 2006*). These plasma GH profiles, in turn, regulate sex-specific gene transcription in the liver through both positive regulatory mechanisms (class I sex-biased genes) and negative regulatory mechanisms (class II sex-biased genes) (*Wauthier et al., 2010*).

The sex-dependent hepatic actions of GH require the transcription factor STAT5 (*Clodfelter et al., 2006*), which is activated by phosphorylation on a single tyrosine residue catalyzed by the GH receptor-associated tyrosine kinase JAK2 (*Waters, 2016*). Each successive male plasma GH pulse induces a cycle of STAT5 tyrosine phosphorylation, dimerization, and nuclear translocation, followed by tyrosine dephosphorylation and recycling of STAT5 back to the cytosol in time to reset the overall signaling pathway for the next plasma GH pulse (*Waxman and Holloway, 2009*). In male mouse liver, GH pulse-activated STAT5 binds to the DNA motif TTCNNNGAA at genomic sites strongly enriched for proximity to male-biased genes (*Zhang et al., 2012*), but the mechanistic relationship between STAT5 binding and sex differences in chromatin accessibility and epigenetic marks is unknown. In female liver, persistent activation of STAT5 by the near-continuous presence of circulating GH is associated with a significant enrichment of STAT5 binding nearby female-biased genes (*Zhang et al., 2012*); however, the underlying mechanisms, including chromatin features that distinguish these STAT5 binding sites from male-biased STAT5 binding sites, are poorly understood.

Liver STAT5 is activated by male plasma GH pulses within minutes, enabling STAT5 to rapidly induce the transcription of several male-biased genes (*Connerney et al., 2017*); however, a majority of male-biased genes respond slowly to the feminization of GH secretory patterns (*Lau-Corona et al., 2017*), which likely reflects the time required for secondary changes, including changes in histone modifications and the underlying chromatin state (*Sugathan and Waxman, 2013*). Thus, continuous infusion of GH in male mice overrides the endogenous plasma GH pulses and substantially feminizes liver gene expression over a period of days, with female-biased genes already in an active chromatin state in male liver often responding earlier than genes in an inactive chromatin state (*Lau-Corona et al., 2017*). Changes in chromatin accessibility occur at genomic regions identified as DNase-I hypersensitive sites (DHS), a hallmark of epigenetic regulation. These DHS can be discovered by DNase-seq, which has identified ~70,000 DHS in male and female mouse liver (*Ling et al., 2010*), including thousands of enhancers, promoters, and insulator regions (*Matthews and Waxman, 2018*). More than 4000 of the 70,000 liver DHS show sex differences in chromatin accessibility, as well as sex-biased binding of STAT5 and other GH-regulated transcription factors (*Sugathan and Waxman, 2013*; *Ling et al., 2010*). These transcription factors reinforce sex differences in liver gene expression and are key regulators of downstream sex differences in disease susceptibility (*Waxman and Kineman, 2022*; *Nikkanen et al., 2022*; *Baik et al., 2011*). Importantly, male-biased transcription factor binding is strongly enriched at male-biased DHS located nearby male-biased genes, and female-biased transcription factor binding is strongly enriched at female-biased DHS found nearby female-biased genes. Sex differences in chromatin structure and accessibility are thus key features of sex-differential liver gene expression (*Lau-Corona et al., 2017*; *Sugathan and Waxman, 2013*). However, little is understood about the mechanisms linking the sex-dependent temporal GH secretion patterns to the robust sex differences in chromatin accessibility and transcription factor binding that regulate liver gene expression.

Here, we elucidate the relationship between the sex-dependent patterns of plasma GH stimulation of hepatocytes and sex differences in liver chromatin accessibility. We identify more than 800 male-biased enhancer DHS regions, where the direct binding of plasma GH pulse-activated STAT5 induces a dynamic cycle of male liver chromatin opening and closing at sites that comprise 31% of all male-biased DHS. Thus, the pulsatility of plasma GH stimulation per se confers significant male bias in chromatin accessibility, and of STAT5 binding, at a substantial fraction of the genomic sites linked to male-biased liver gene expression. Furthermore, we establish that a single physiological replacement dose of GH given as a pulse to hypophysectomized (hypox) mice recapitulates, within 30 min, the pulsatile reopening of chromatin seen in pituitary-intact male mouse liver. Pulsatile chromatin opening is thus a novel mechanism controlling sex differences in chromatin accessibility and transcription factor binding closely linked to sex differences in gene expression and liver disease. Further, we elucidate key epigenetic features distinguishing this dynamic, STAT5-driven mechanism of male-biased chromatin opening from that operative at a second, distinct class comprised of static male-biased DHS, which are constitutively open in male liver but closed in female liver. Finally, our analysis of histone marks enriched at each class of male-biased DHS, and at a third sex-biased DHS class, comprised of static female-biased DHS, elucidates distinct epigenetic mechanisms mediate sex-specific gene repression in each sex.

## Results
### Sex-biased DHS are primarily distal enhancers that target sex-biased genes

Open (accessible) chromatin regions identified in mouse liver by DNase-seq analysis have been classified as enhancers, insulators, and promoters based on their chromatin mark patterns and CTCF binding activities (n = 70,211 standard reference DHS set) (*Matthews and Waxman, 2018*; *Supplementary file 1A*). Several thousand of these DHS show greater chromatin accessibility in male than female liver (n = 2729 male-biased DHS) or greater accessibility in female than male liver (n = 1366 female-biased DHS) (*Ling et al., 2010*). These sex-biased DHS regions were enriched for enhancer histone marks, with >85% classified as enhancers or weak enhancers in mouse liver, compared to 66% for all sex-independent DHS (*Figure 1A*). Assignment of the sex-biased liver DHS to their putative gene targets (closest RefSeq gene or multi-exonic lncRNA gene transcription start site in the same topologically associating domain [TAD]) (*Matthews and Waxman, 2018*) revealed that sex-biased DHS were highly enriched for mapping to genes showing a corresponding sex bias in their level of transcription, but not for genes whose expression shows the opposite sex bias (*Supplementary files 1B and 2*). Cumulative plots of the distance from each DHS to its target gene revealed that a majority of sex-biased DHS classified as enhancers or insulators are distal to their target genes, whereas sex-biased DHS with H3K4me3 marks, which comprise only 1–2% of all sex-biased DHS, are proximal to their target genes, validating their classification as promoter DHS (*Figure 1B*). Thus, a majority of sex-biased DHS have the marks of positively acting distal enhancers, consistent with our finding that sex-dependent DNA looping at an intra-TAD scale is common in mouse liver (*Matthews and Waxman, 2020*).

No major differences in chromatin state distributions between male-biased, female-biased, and sex-independent enhancer DHS were seen based on chromatin state maps developed separately for male and female mouse liver (*Supplementary file 3*; *Sugathan and Waxman, 2013*). Thus, all three sets of enhancer DHS showed a high frequency (50–66%) of chromatin state E6, whose emission probabilities (*Figure 1C*) indicate a high frequency of DHS in combination with the activating chromatin marks H3K27ac and H3K4me1, and a lower frequency (12–16%) of chromatin state E5 (high frequency of DHS but low frequency of H3K27ac and H3K4me1) (*Figure 1D*). Sex-biased and sex-independent insulator DHS also showed similar chromatin state distributions (state E5 ~ state E6), as did sex-biased and sex-independent promoter DHS, which were primarily in state E7, characterized by a high frequency of H3K4me3 and H3K4me1 marks (*Figure 1D*).

The absence of major differences in overall chromatin state distributions between male-biased and female-biased liver DHS prompted us to investigate other factors that may provide insight into the underlying mechanisms regulating sex differences in hepatic chromatin accessibility and their link to sex differences in gene expression.

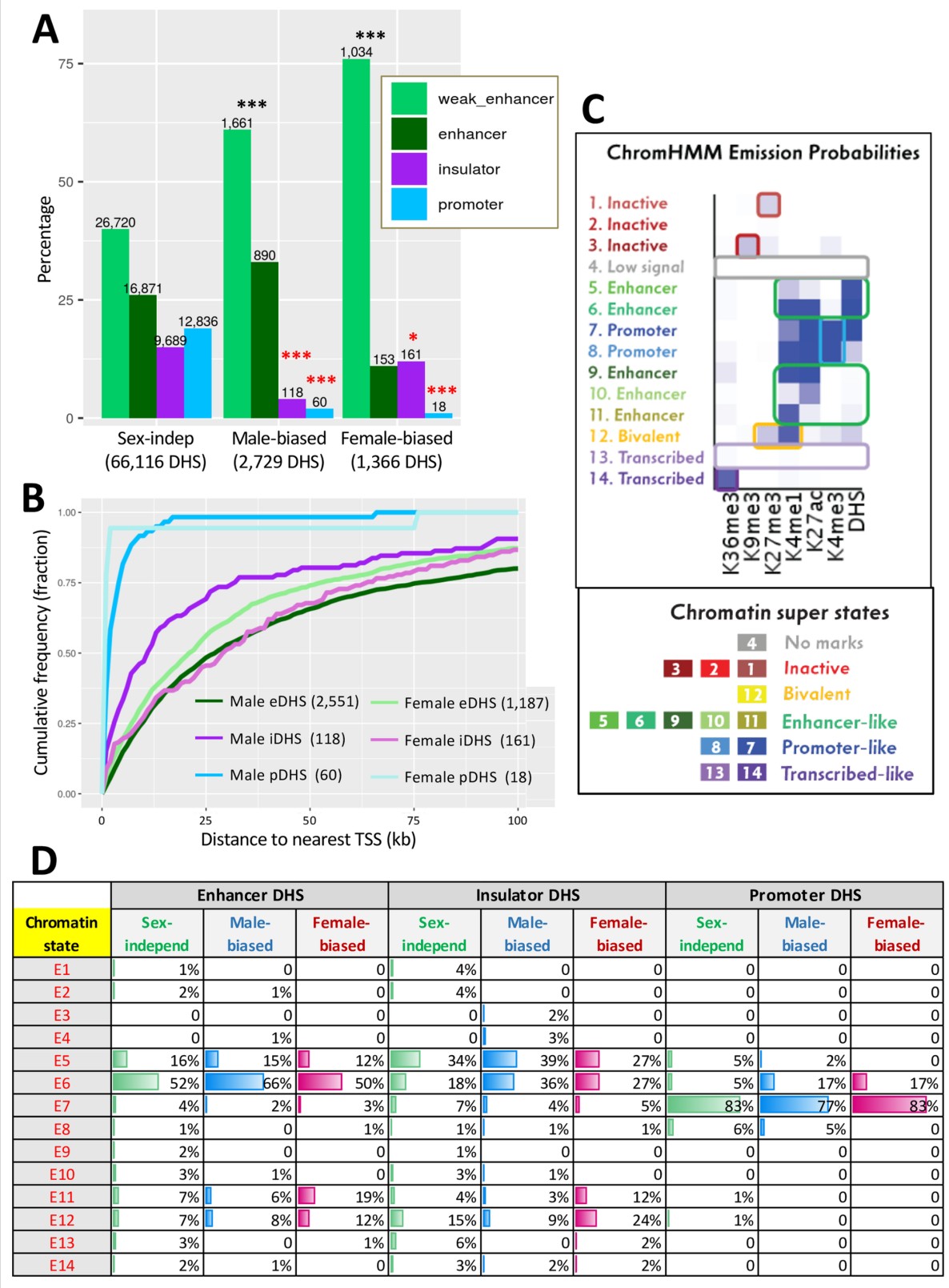

**Figure 1.** Liver DNase-I hypersensitivity site (DHS) classification and mapping to liver-expressed genes. (**A**) Distributions of DHS classified as weak enhancer, enhancer, insulator, or promoter (*Matthews and Waxman, 2018*) for the indicated three DHS sets. Values above bars, number of DHS. Enrichments of sex-biased DHS for being an enhancer or weak enhancer, an insulator, or a promoter DHS were determined by comparing to a background set of sex-independent DHS (66,116 sites); significance was determined by Fisher's exact test with Benjamini–Hochberg p-value adjustment:

*Figure 1 continued on next page*

*Figure 1 continued*

*p<0.01; **p<1E-10; ***p<1E-50. Black asterisks, enrichment; red asterisks, depletion as compared to background DHS set. (**B**) Cumulative frequency distribution of the distance to the nearest transcription start site in the same topologically associating domain (TAD) for male-biased and female-biased enhancer (e), insulator (i), and promoter (p) DHS. (**C**) ChromHMM emission probabilities for each of the 14 chromatin states developed for male and female mouse liver (*Sugathan and Waxman, 2013*), which serves as a reference for data shown in panel (**D**) and in *Figure 7*. Summary descriptions of the characteristics of each state are shown at the left and below. (**D**) Chromatin state distributions for each sex-biased or sex-independent DHS set. Chromatin state data for male and female liver (*Supplementary file 3*) was used for male-biased and female-biased DHS, respectively (*Supplementary file 1D*).

## Impact of endogenous pulses of GH and STAT5 activity in male liver

Sex differences in pituitary GH secretion – pulsatile in males vs. near continuous (persistent) in females – regulate the sex-dependent expression of hundreds of genes in adult mouse liver. This regulation requires the GH-activated transcription factor STAT5 (*Hao and Waxman, 2021*; *Holloway et al., 2007*). We hypothesize that the pulsatile activation of STAT5 seen in male liver in direct response to plasma GH stimulation (*Zhang et al., 2012*; *Connerney et al., 2017*; *Tannenbaum et al., 2001*) dynamically alters the chromatin accessibility landscape of male mouse liver. More specifically, we propose that the repeated activation of STAT5 in male mouse liver by plasma GH pulses induces dynamic cycles of chromatin opening and closing at a subset of liver DHS, and that this response can be discovered by comparing chromatin accessibility profiles in livers from individual male mice euthanized at a peak vs. at a trough of hepatic STAT5 activity (*Figure 2A*, top). To test this hypothesis, we analyzed DNase-seq libraries prepared from liver nuclei purified from 21 individual adult male mice (*Supplementary file 4*). In parallel, we determined the STAT5 DNA-binding activity of each liver by electrophoretic mobility shift analysis (EMSA) of whole-liver extracts. Of the 21 livers, 10 had high STAT5 EMSA activity (STAT5-high-activity livers) and 11 had very low or no detectable STAT5 EMSA activity (STAT5-low-activity livers) (*Figure 2B*, *Figure 2—figure supplement 1*). Next, we performed DNase-seq analysis on genomic DNA fragments released by light DNase-I digestion of nuclei purified from each liver, followed by diffReps analysis (*Shen et al., 2013*) comparing the DNase-seq-released DNA fragments from each group of livers. We thus discovered genomic regions showing significant differential chromatin accessibility between STAT5-high and STAT5-low male livers. Principal component analysis using either the top 200 or the top 600 most significant diffReps-identified differentially accessible regions revealed that 18 of the 21 livers gave patterns of DNase-released fragments that correlate with STAT5 activity. Two of the STAT5-high livers and one of the STAT5-low livers were outliers (*Figure 2C*). These same three outliers could also be identified by their discordant DNase-seq read count distributions (*Figure 2D*) and were excluded from all downstream analyses.

We reanalyzed the remaining set of 18 STAT5-high and STAT5-low livers using diffReps, and then filtered the output differential peak list to retain those sites identified as DHS peaks by MACS2 analysis of the same 18 DNase-seq datasets. We thus identified n = 2832 genomic sites where chromatin opening is associated with high liver STAT5 activity and n = 123 other sites where chromatin opening is associated with low liver STAT5 activity (*Supplementary file 5A*). As each liver represents a time point of peak (STAT5-high livers) or trough (STAT5-low livers) levels of GH pulse-activated liver STAT5 activity (*Figure 2A*), the STAT5-high/STAT5-low differential sites identify genomic regions where chromatin dynamically opens or closes in male mouse liver in close association with GH pulse activation of STAT5. The greater chromatin opening in STAT5-high livers compared to STAT5-low male livers (and compared to female livers) was visualized in aggregate plots of normalized DNase-seq cuts across the 2832 genomic regions (*Figure 2E*, left; *Supplementary file 6*). Chromatin accessibility was greater in the STAT5-low livers at the 123 STAT5-low diffReps-identified genomic sites, where STAT5 activation is associated with chromatin closing (*Figure 2E*, right). This pattern, where individual male mouse livers largely show either high or low DNase-seq read count distributions at the top differential genomic sites, was also seen in an independent set of nine male liver DNase-seq samples from a second mouse strain generated by the ENCODE consortium: four of the nine livers showed normalized sequence read distributions very similar to the STAT5-high livers, while five of the nine livers were similar to the STAT5-low livers (*Figure 2D*, green bars).

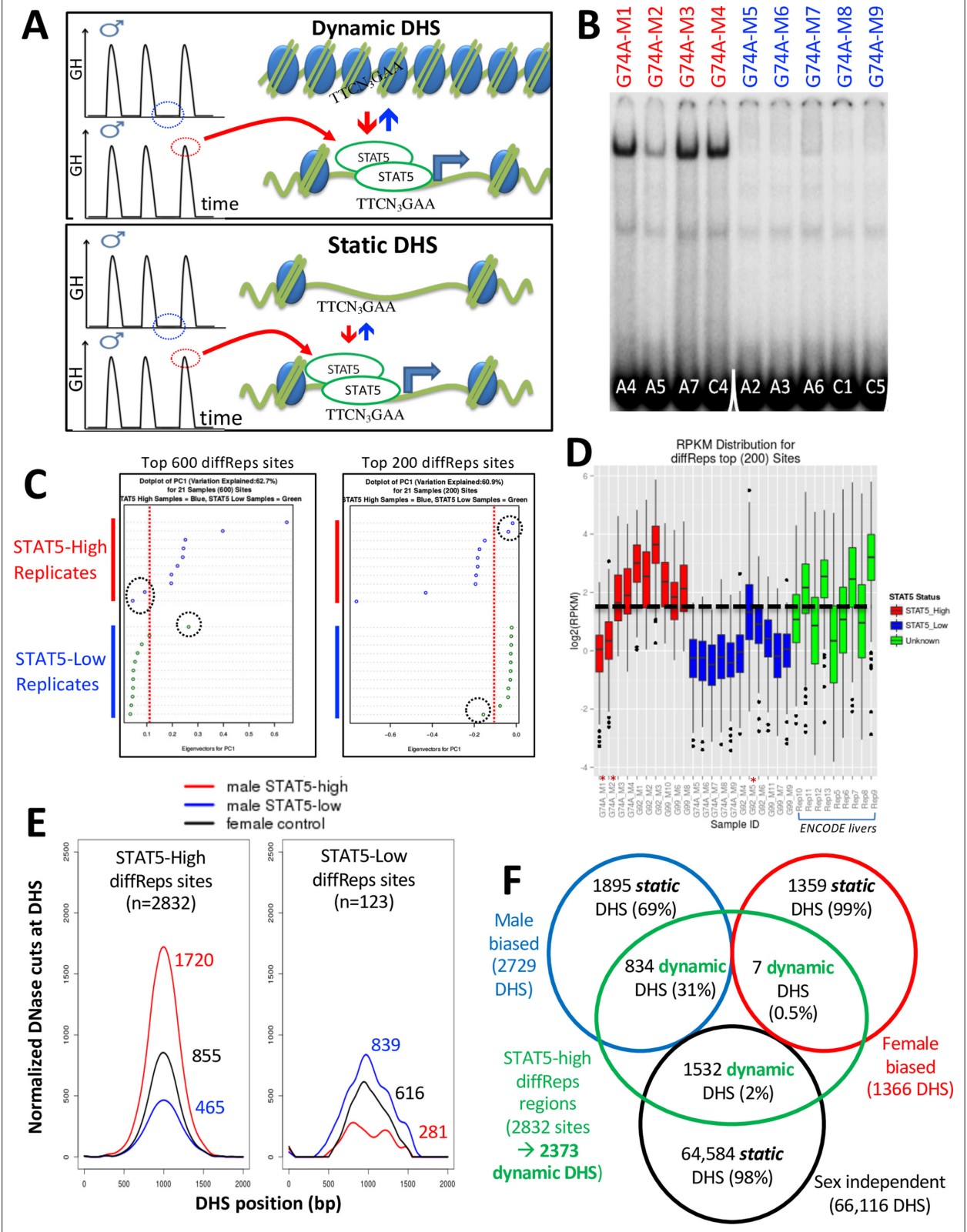

**Figure 2.** Discovery and characterization of dynamic and static male-biased DNase-I hypersensitivity site (DHS). (**A**) Model showing pulsatile male plasma growth hormone (GH) pattern, with mice sampled between GH pulses, when STAT5 is inactive and cytoplasmic (STAT5-low, blue), or at a peak of plasma GH, when liver STAT5 is activated to its homodimeric, nuclear DNA-binding form, which enables STAT5 to open chromatin and bind to its consensus motif, TTCNNNGAA (STAT5-high, red). This intermittent (pulsatile) activation STAT5 leads to chromatin opening and closing at dynamic DHS

*Figure 2 continued on next page*

*Figure 2 continued*

(top). Static DHS also bind STAT5 intermittently (*Zhang et al., 2012*) but remain open between plasma GH pulses (bottom). (**B**) Electrophoretic mobility shift analysis (EMSA) of STAT5 DNA-binding activity in liver extracts prepared from individual male mice. These data represent liver extracts from one of three separate cohorts of mice; the other two cohorts are shown in *Figure 2—figure supplement 1*. Labels at the top indicate the DNase-seq library ID for each liver sample (*Supplementary file 4*). Red labels at the top indicate STAT5-high activity based on the EMSA patterns displayed, and blue labels indicate STAT5-low-activity livers. Numbers at the bottom: mouse ID #. Samples were all run on the same gel. (**C**) Principal component (PC) analysis of the distributions of DNase-seq reads per kilobase per million mapped reads for the top 600 or top 200 diffReps-identified sites that are more open in STAT5-high compared to STAT5-low livers. Eigenvector values for principal component 1 are shown for the individual STAT5-high and STAT5-low liver samples. Dotted red line: empirical cutoff separating STAT5-high from STAT5-low DNase-seq samples; dotted black circles: outlier samples in each dataset. (**D**) Boxplots of DNase-seq activity, in log2(reads per kilobase per million mapped reads), across the top 200 diffReps differential sites that open (as in **C**) for the STAT5-high DNase-seq libraries (red bars), for the STAT5-low DNase-seq libraries (blue bars), and for DNase-seq libraries for nine individual male mouse liver ENCODE consortium samples (green bars, replicates 5–13, marked on x-axis). Thick dashed black line: empirical cutoff used to separate STAT5-high and STAT5-low liver samples. Liver samples that did not pass the cutoff (red asterisks at the bottom) are the same outliers circled in panel (**C**). (**E**) Normalized DNase-I cut site aggregate plots for STAT5-high (red) and STAT5-low male livers (blue), and for female livers (black). Peak cut site y-axis values are shown to the right of each peak. Cut sites were aggregated across the sets of diffReps-identified DHS that show greater diffReps normalized DNase-seq signal intensity in STAT5-high compared to STAT5-low male livers (left), or vice versa (right), that is, that open or close, respectively, in response to endogenous STAT5 pulses in male liver. (**F**) Venn diagram indicating overlap between endogenous STAT5 pulse-opened DHS sets identified by diffReps (2832 sites that respond to liver STAT5 activity in a dynamic manner, which map to a total of 2373 of the 70,211 standard reference DHS) and the indicated sets of sex-biased and sex-independent DHS that do not respond to a change in liver STAT5 activity, that is, are static DHS. See *Supplementary file 1A*, column I for full listing.

The online version of this article includes the following figure supplement(s) for figure 2:

**Figure supplement 1.** Electrophoretic mobility shift analysis (EMSA) of STAT5 DNA-binding activity in liver extracts prepared from individual male mice.

**Figure supplement 2.** Top enriched Gene Ontology (GO) terms identified by GREAT analysis of the predicted gene targets of each of the indicated four DNase-I hypersensitivity site (DHS) sets, shown in panels **A–D**.

## Dynamic vs. static liver DHS

A comparison of the set of 2832 STAT5-high genomic sites with our standard reference set of 70,211 liver DHS revealed that 31% (n = 834) of the 2729 male-biased liver DHS described above are STAT5-high sites, that is, they respond dynamically to the pulsatile activation of liver STAT5 by endogenous male plasma GH pulses. In contrast, only 0.5% of female-biased liver DHS and 2% of sex-independent liver DHS showed this dynamic response to STAT5 (*Figure 2F*, green). This strong enrichment of GH/STAT5 pulse-induced chromatin opening at male-biased DHS compared to sex-independent DHS (ES = 12.9; p<1E-05) identifies intermittent chromatin opening induced by male plasma GH pulses as a mechanism that can explain the male bias in chromatin accessibility for a significant subset (31%) of male-biased DHS.

To determine whether STAT5 binding per se is an important feature of the observed pulsatile changes in chromatin accessibility at these sites in male liver, we examined normalized DNase-I cut site aggregate plots for the subset comprised of n = 1307 male-biased DHS that bind STAT5 in ChIP-seq analyses (*Zhang et al., 2012*). STAT5-high livers showed the highest mean level of chromatin opening, followed by up to a 2.8-fold lower level of chromatin opening at those same genomic regions in STAT5-low male livers and in female livers (*Figure 3A*, plot 1). In contrast, male-biased DHS that did not bind STAT5 (n = 1422 DHS) showed nearly equal chromatin accessibility in STAT5-high vs. STAT5-low male livers but approximately twofold lower accessibility in female livers (*Figure 3A*, plot 2). Thus, the male bias in accessibility at the STAT5-bound but not at the non-STAT5-bound male-biased DHS can be explained by STAT5-induced pulsatile chromatin opening in male liver. It is also apparent that the male bias of the non-STAT5-bound DHS set is associated with chromatin closing in female liver (*Figure 3A*, plot 2, black). The conclusion that chromatin is relatively closed at these 1422 DHS in female liver is also evident by comparing their mean normalized DNase cutting frequency to that of the full genome-wide set of 53,404 STAT5-unbound, sex-independent DHS (*Figure 3A*, plot 2, black, vs. plot 4). Chromatin accessibility was much higher at the subset of sex-independent DHS that bound STAT5 (n = 12,712 DHS, plot 3). Importantly, the high chromatin accessibility at these sex-independent DHS was seen in livers of both sexes and was largely invariant between STAT5-high and STAT5-low male livers. We conclude that STAT5 binding is associated with an open chromatin state, and that STAT5-associated dynamic chromatin opening and closing is a defining characteristic of a specific subset of male-biased DHS but occurs infrequently at female-biased and sex-independent DHS.

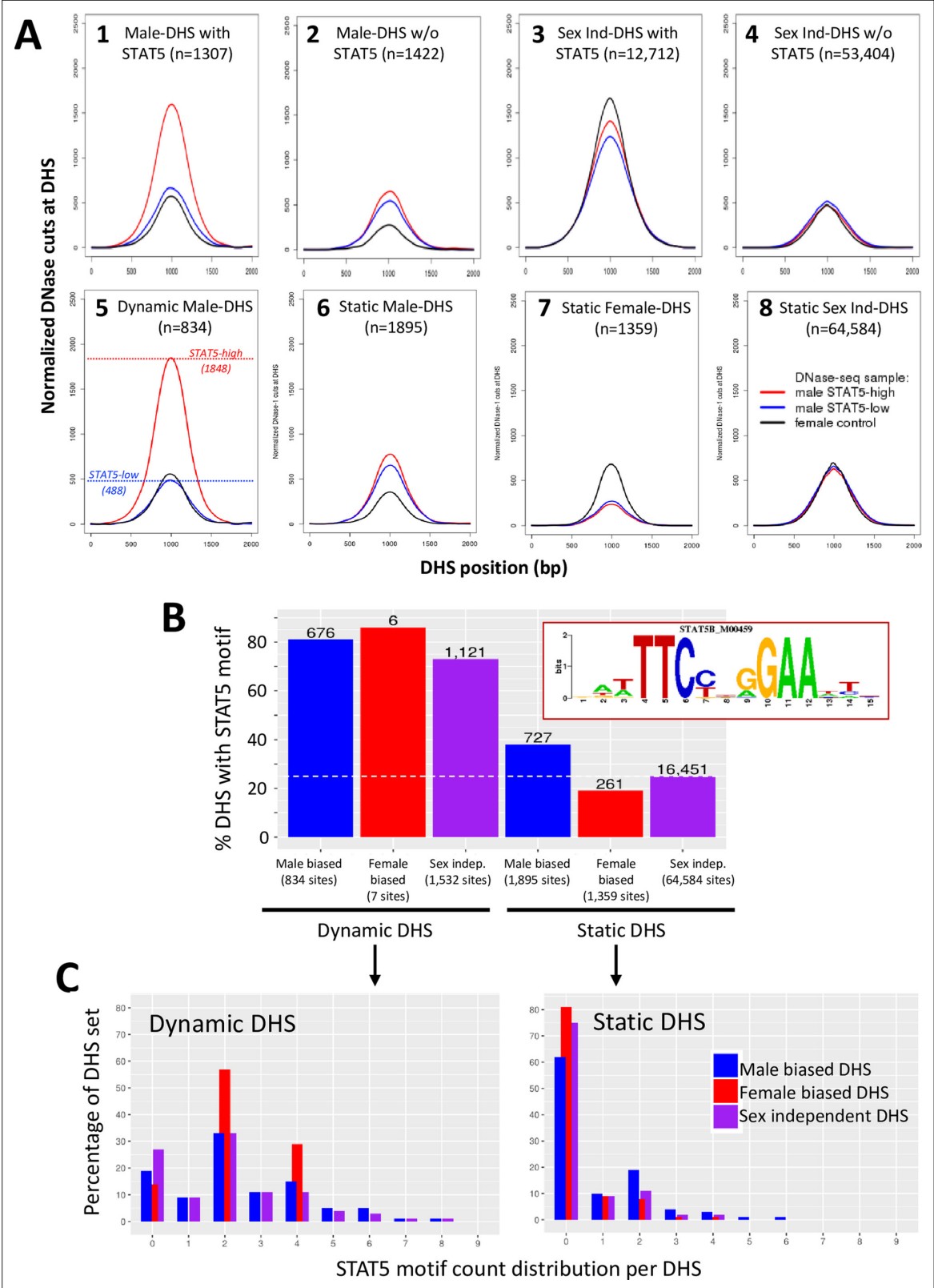

**Figure 3.** Dynamic and static male-biased DNase-I hypersensitivity sites (DHS). (**A**) Normalized DNase-I cut site aggregate plots for STAT5-high (red) and STAT5-low male livers (blue) and female livers (black) across the genomic regions included in the sets of male-biased DHS (plots 1 and 2) and sex-independent DHS (plots 3 and 4), separated into subsets of DHS that either do (plots 1 and 3) or do not bind STAT5 by ChIP-seq analysis (plots 2 and 4). Plots 5–8 show corresponding plots for each of the indicated dynamic and static DHS sets. See *Supplementary file 6* for normalized peak DNase-I site

*Figure 3 continued on next page*

*Figure 3 continued*

values. (**B**) Bar plots showing number (values above bars) and percent of static and dynamic DHS with one or more occurrences of the indicated STAT5 motif (box), based on FIMO scan of the 70,211 standard reference DHS sequences. Dashed horizontal line: background, genome-wide occurrence of STAT5 motifs at static sex-independent DHS. (**C**) Percentage of dynamic (left) and static (right) DHS sets with zero or more occurrences of a STAT5 motif. x-axis: number of motif occurrences in each individual DHS (n = 0–9); y-axis: percentage of full DHS set with the corresponding number of STAT5 motifs. Distribution pattern for dynamic female-biased DHS is not reliable, as it represents a total of only seven DHS.

The online version of this article includes the following figure supplement(s) for figure 3:

**Figure supplement 1.** STAT5 binding is associated with chromatin opening.

**Figure supplement 2.** TF motifs discovered de novo from sets of DNase-I hypersensitivity site (DHS) sequences, part 1.

**Figure supplement 3.** TF motifs discovered de novo from sets of DNase-I hypersensitivity site (DHS) sequences, part 2.

The above findings allow us to define two distinct subsets of male-biased DHS, which differ in their responsiveness to liver STAT5 activation: (1) *dynamic male-biased DHS* are characterized by a male bias in chromatin accessibility linked to an increase in chromatin opening following the binding of STAT5 when activated by a plasma GH pulse in male liver every 3–4 hr (*Waxman and Holloway, 2009*; *Connerney et al., 2017*) (n = 834 dynamic male-biased DHS, 31%); and (2) *static male-biased DHS* are open in male liver constitutively, that is, throughout the pulsatile, on/off cycles of GH-induced STAT5 activity in male liver, and are comparatively closed in female liver (n = 1895 static male-biased DHS, 69%). Supporting this conclusion, mean chromatin opening at the set of 834 dynamic male-biased DHS was 3.8-fold higher in STAT5-high male livers than in STAT5-low male livers or in female livers (*Figure 3A*, plot 5). In contrast, the higher chromatin accessibility in male than female liver at the set of 1895 static male-biased DHS was largely independent of the male liver's STAT5 activity status

**Table 1.** Sex-biased DNase-I hypersensitivity sites (DHS).

| | Dynamic male-biased DHS (n = 834, 31%) | Static male-biased DHS (n = 1895, 69%) | Static female-biased DHS (n = 1359, 99%) |
|---|---|---|---|
| *A. DHS activity (extent of chromatin opening)* | | | |
| STAT5-high male liver | +++ | + | - |
| STAT5-low male liver | + | + | - |
| Female liver | + | - | + |
| *B. Sex-biased TF binding sites (enrichment)* | | | |
| STAT5 (binding in male liver) | +++ | + | - |
| CUX2 (binding in female liver) | + | ++ | - |
| FOXA1 (male-biased sites) | + | ++ | - |
| FOXA2 (male-biased sites) | +++ | +++++ | - |
| FOXA2 (female-biased sites) | - | - | +++ |
| *C. Enriched H3 histone marks and chromatin states (enrichment)* | | | |
| K27ac, K4me1, with DHS (male liver) [State E6] | ++ | ++ | - |
| K27ac and/or K4me1 (male-liver) [States E10, E11] | - | - | ++ |
| K36me3 (male-biased) | - | ++ | - |
| K36me3 (female-biased) | - | - | ++ |
| K27me3 (male-biased) | - | - | +++ |
| K9me3 (female-biased) | ++ | ++ | - |

+++, high; ++, medium; +, low; -, very low or absent.

(*Figure 3A*, plot 6). Of note, the mean level of male liver chromatin opening at those 1895 sites was 2.6-fold lower than the peak level of accessibility seen at the dynamic male-biased DHS regions in STAT5-high male livers (*Figure 3A*, plot 6 vs. 5; *Table 1A*).

The mean level of chromatin accessibility at the set of static female-biased DHS (1359 sites, *Figure 2F*) was 2.7–2.9-fold higher in female liver than in male liver, where accessibility was independent of male plasma GH/STAT5 pulses (*Figure 3A*, plot 7). Overall, the accessibility of the static female-biased DHS in female liver was very similar to that of the genome-wide set of 64,584 static sex-independent DHS (*Figure 3A*, plot 8 vs. 7; peak normalized DNase-I activity value of 699 vs. 683, *Supplementary file 6*).

## STAT5 binding is closely associated with dynamic male-biased DHS

The presence of a canonical STAT5 motif, TTCNNNGAA, is a distinguishing feature of dynamic male-biased DHS: 81% of the 834 dynamic male-biased DHS contained a STAT5 motif vs. only 38% of the 1895 static male-biased DHS (c.f., 25% background motif frequency at 64,584 sex-independent static DHS) (*Figure 3B*). In addition, multiple STAT5 motifs are more frequently found at dynamic DHS than at static DHS (*Figure 3C*). Consistent with this, STAT5 binding, determined experimentally by ChIP-seq, occurs at a greater fraction of dynamic than static male-biased DHS (85% vs. 32%; *Figure 3—figure supplement 1A*), and the level of STAT5 binding (normalized STAT5 ChIP-seq read counts) was significantly higher at the STAT5-bound subsets of the dynamic vs. static DHS sets (*Figure 3—figure supplement 1B*). De novo motif discovery supported these findings, with top-scoring motifs matching the STAT5B motif found in the dynamic DHS but not in the static DHS sets (*Figure 3—figure supplement 2* vs. *Figure 3—figure supplement 3*). A close association between STAT5 binding and chromatin opening is also indicated by the higher chromatin accessibility at the DHS subsets associated with STAT5 binding (*Figure 3—figure supplement 1C*, top row vs. bottom row).

These findings establish a close link between STAT5 binding, which is pulsatile in male liver (*Zhang et al., 2012*), and the repeated opening and closing of chromatin at dynamic DHS. Nevertheless, pulsatile chromatin opening was also found at 124 male-biased DHS (15% of all dynamic male-biased DHS) that did not bind STAT5 (*Figure 3—figure supplement 1C*, plot 1B). Moreover, STAT5 binding alone is not sufficient to ensure dynamic, male-biased chromatin opening and closing, insofar as 90% of all sex-independent DHS that bind STAT5 do not undergo dynamic chromatin opening and closing (*Figure 3—figure supplement 1C*, plot 5A [11,473 sites, 90.3%] vs. 4A [1239 sites, 9.7%]). Furthermore, only 54% of male-biased DHS that bind STAT5 are dynamic DHS (710 sites), the other 46% (597 sites) being static male-biased DHS, even though they bind STAT5 (*Figure 3—figure supplement 1C*, plot 1A vs. 2A), although the level of STAT5 binding is lower than that at dynamic male-biased DHS and is similar to the STAT5-bound static sex-independent DHS (*Figure 3—figure supplement 1B*). Thus, factors other than pulsatile STAT5 binding per se are required for dynamic chromatin opening and closing to occur. Finally, dynamic DHS showed a strong preference for male-biased chromatin accessibility, with the occurrence of STAT5-associated dynamic chromatin opening being 5.6-fold greater at the STAT5-bound male-biased DHS (54%) than at the STAT5-bound sex-independent DHS (9.7%).

## Enrichment of other GH-regulated liver transcription factors at dynamic vs. static DHS

We used published ChIP-seq data to investigate whether dynamic and static male-biased DHS differ with respect to the binding of other, sex-biased transcription factors (*Table 1*, *Supplementary file 7A*). We examined two GH/STAT5-regulated transcriptional repressors that reinforce STAT5 regulation of liver sex differences. One factor, BCL6, is a male-biased protein that can compete for STAT5 binding to chromatin and preferentially represses expression of female-biased genes in male liver (*Zhang et al., 2012*; *Sugathan and Waxman, 2013*; *Nikkanen et al., 2022*; *Meyer et al., 2009*). The second factor, CUX2, is a female-specific repressor protein whose binding sites in female liver are enriched nearby male-biased genes, which enables CUX2 to repress those genes in female liver (*Conforto et al., 2012*; *Conforto et al., 2015*). Both repressors showed significant enrichment for binding to genomic regions defined by male-biased DHS compared to a background set of static sex-independent DHS. Thus, male-biased BCL6 binding was significantly enriched at the set of dynamic male-biased DHS (ES = 2.0, p=E-14) but showed minimal enrichment at static male-biased DHS (ES

= 1.3, p=1.6E-04). BCL6 binding was also enriched at dynamic sex-independent DHS (*Supplementary file 7B*). In contrast, CUX2 was most highly enriched for binding in female liver at genomic sites that correspond to static male-biased DHS (ES = 4.6, p=E-40), and de novo motif discovery identified a CUX2 motif as the top enriched motif in this, but not the other DHS sets (*Figure 3—figure supplement 2C*). This binding of CUX2 may contribute to the greater closure of those DHS in female compared to male liver (*Figure 3A*, plot 6). Finally, male-biased STAT5 binding sites showed 2.5-fold greater enrichment at dynamic than at static male-biased DHS (ES = 58, p=E-301 vs. ES = 24, p=E-190) (*Supplementary file 7B*), consistent with our findings, above, implicating pulsatile STAT5 binding in dynamic chromatin opening at those sites.

We reanalyzed published mouse liver ChIP-seq data for FOXA1 and FOXA2 (Array Express Biostudies accession # E-MTAB-805) (*Li et al., 2012*), pioneer factors implicated in chromatin opening (*Balsalobre and Drouin, 2022*; *Zaret, 2020*), to identify sex-dependent binding sites for each factor. Male-biased binding sites discovered for each FOXA factor showed strong enrichment for binding at male-biased DHS, with up to 2-fold higher enrichments seen at the static male-biased DHS (for FOXA1: ES = 6.1 [static DHS] vs. ES = 3.7 [dynamic DHS]; for FOXA2: ES = 44 [static DHS] vs. ES = 21 [dynamic DHS]) (*Table 1*, *Supplementary files 7B and 8*). Consistent with this, de novo motif discovery identified a Fox family factor, FOXI1, as a close match for one of the top enriched motifs in the set of static but not in the set of dynamic male-biased DHS (*Figure 3—figure supplement 2*, C vs. A; p-value 9.2E-06 vs. 2.0E-03). Finally, female-biased FOXA2 binding sites, but not female-biased FOXA1 binding sites, showed strong enrichment for static female-biased DHS (ES = 18, p=E-75; *Supplementary file 7B*). Taken together, these findings support the proposal that FOXA2, and to a lesser extent FOXA1, contribute to sex-dependent chromatin opening, in particular at static DHS.

## Impact of hypophysectomy on liver chromatin accessibility

Surgical removal of the pituitary gland (hypophysectomy) ablates pituitary GH secretion and thereby abolishes liver STAT5 activation (*Connerney et al., 2017*), leading to widespread loss of sex-specific liver gene expression (*Wauthier et al., 2010*). We hypothesized that by ablating GH-induced STAT5 activation and DNA binding, hypophysectomy will lead to closure of many of the open chromatin regions that regulate sex-specific gene expression. Furthermore, we proposed that restoration of a pulsatile GH signal, in the form of a single exogenous GH pulse given by i.p. injection, will reopen chromatin at many of those sites (*Figure 4A*). To test this hypothesis, we used DNase-seq to compare the chromatin accessibility profiles of liver nuclei from hypox male and hypox female mice to those of pituitary-intact control liver nuclei. Hypophysectomy induced changes in chromatin accessibility at several thousand sites, including large numbers of sex-independent DHS, with many more genomic regions undergoing chromatin closing than chromatin opening in male liver, but not in female liver (*Figure 4B*, *Supplementary file 5*). Importantly, male-biased DHS that responded to hypophysectomy were almost exclusively closed in male liver following hypophysectomy (*Figure 4C*, first two rows, % opening vs. % closing). A much smaller percentage of static female-biased DHS responded to hypophysectomy, and the responses observed generally involved chromatin closing in female liver and chromatin opening in male liver (*Figure 4C*). Thus, sex-biased DHS are subject to both positive and negative pituitary hormone regulation and respond differently between the sexes.

## DHS responses to pituitary ablation link to class I and II sex-biased gene regulatory responses

Hypophysectomy abolishes GH-regulated liver sex differences via two distinct mechanisms, which identify two classes of sex-biased genes (*Wauthier et al., 2010*): hypophysectomy leads to downregulation of class I sex-biased genes in the sex where the gene is more highly expressed, and it leads to upregulation of class II sex-biased genes in the sex where the gene shows lower expression prior to hypophysectomy (*Figure 4—figure supplement 1A*, model). Building on this classification, we hypothesized that DHS that close following hypophysectomy are enhancers mapping to sex-biased genes subject to either positive regulation by GH (class I sex-biased genes) or negative regulation by GH (class II sex-biased genes). Supporting this proposal, the DHS that close in male liver following hypophysectomy showed strong, specific enrichment (ES = 5.7, p=1.3E-69) for mapping to class I male-specific genes, which are downregulated in hypox male liver, whereas the DHS that close in hypox female liver showed strong, specific enrichment (ES = 7.8, p=6.6E-46) for mapping to class I

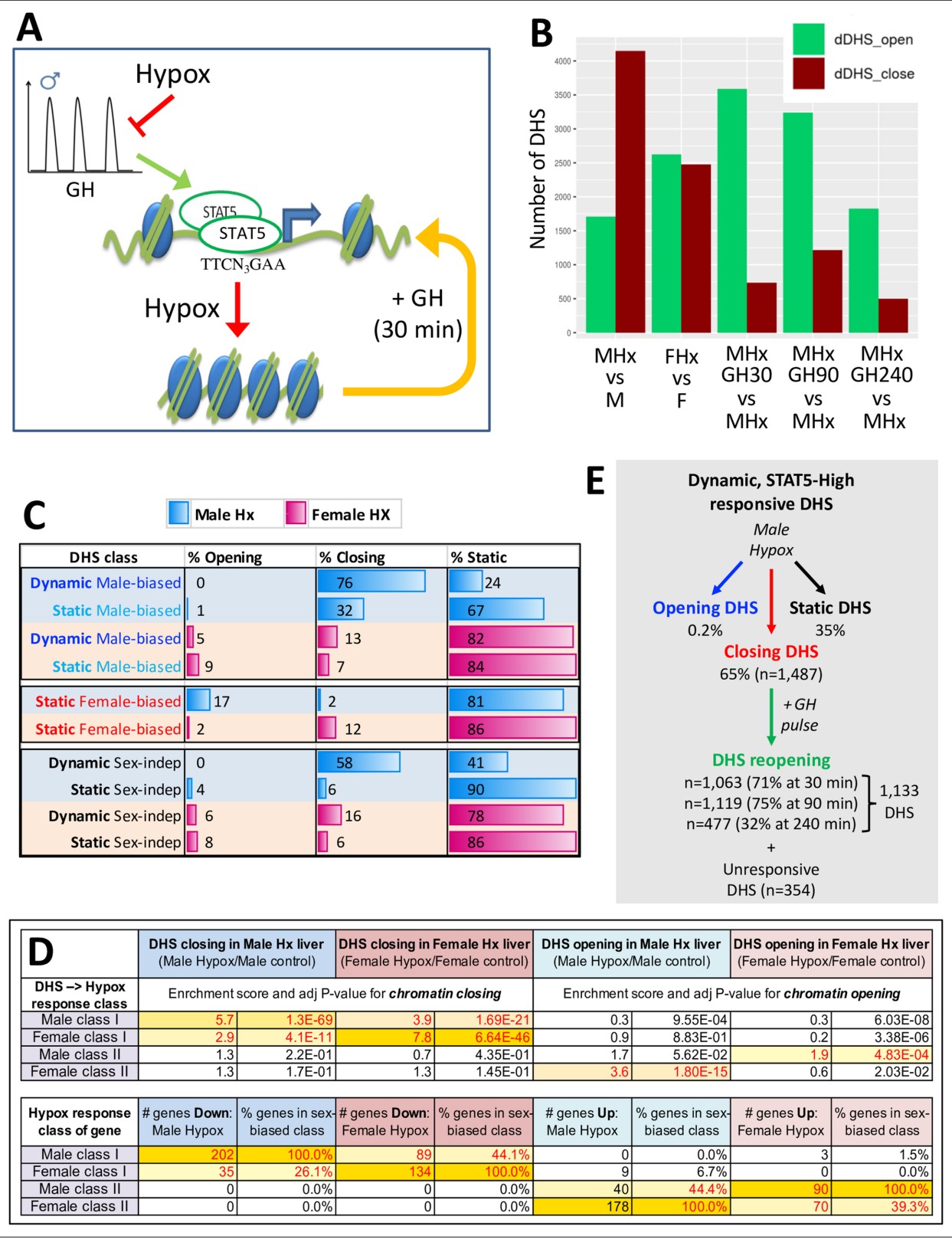

**Figure 4.** DNase-I hypersensitivity sites (DHS) responsive to hypophysectomy, their enrichment for class I and II sex-biased gene targets, and their responses to growth hormone (GH) pulse replacement. (**A**) Model for the impact of pulsatile GH secretion ablation by hypophysectomy on STAT5-induced chromatin opening. DHS that close following hypophysectomy due to the loss of active, DNA-binding STAT5 reopen within 30 min of exogenous GH treatment, which rapidly reactivates STAT5 and induces its nuclear translocation (*Connerney et al., 2017*). (**B**) Numbers of liver DHS that

*Figure 4 continued*

open or close following hypophysectomy (Hx) and in response to a single injection of GH given to hypox male (M) or female (F) mice and euthanized 30, 90, or 240 min later. DHS opening and closing was calculated as compared to the indicated controls. (**C**) Distributions of sex-biased and sex-independent DHS that open, close, or are unchanged (static) following hypophysectomy (Hx) in male and female mouse liver. See ***Supplementary file 5H*** for full details. (**D**) Enrichment of hypophysectomy-responsive DHS for mapping to the four indicated classes of sex-biased genes when compared to a background set of DHS whose accessibility is unchanged by hypophysectomy (see ***Figure 4—figure supplement 1***). Class I and II sex-biased genes were identified from RNA-seq gene expression data collected from intact and hypox male and female liver samples (***Supplementary file 2***). The bottom section shows the total number and percentage of sex-biased genes in each of the four indicated sex-biased gene classes that respond to hypophysectomy, as marked (***Supplementary file 5A and I***). (**E**) Subset of all dynamic DHS (***Figure 2F***) that close following hypophysectomy in male mouse liver (n = 1487) and then respond to GH pulse replacement at the three indicated time points. Total number of dynamic DHS shown here is lower than the full set of 2373 dynamic DHS (***Figure 2F***) as 70 of these DHS were not identified as DHS in the hypophysectomy study (***Supplementary file 5H***).

The online version of this article includes the following figure supplement(s) for figure 4:

**Figure supplement 1.** Sex-biased genes can be classified based on their pituitary hormone dependence.

**Figure supplement 2.** Impact of hypophysectomy and growth hormone (GH) pulse replacement on a set of 2373 dynamic DNase-I hypersensitivity sites (DHS), part 1.

**Figure supplement 3.** Impact of hypophysectomy and growth hormone (GH) pulse replacement on a set of 2373 dynamic DNase-I hypersensitivity sites (DHS), part 2.

female-specific genes, which are downregulated in hypox female liver (***Figure 4D***). In contrast, the DHS that open in hypox male liver were specifically enriched for mapping to class II female-specific genes (ES = 3.6, p=1.8E-15), which are upregulated (de-repressed) in male liver following hypophysectomy, whereas the DHS that open in hypox female liver were specifically enriched albeit only moderately (ES = 1.9, p=4.8E-04) for mapping to class II male-specific genes, which are upregulated in female liver following hypophysectomy (***Figure 4—figure supplement 1B***). The enrichment patterns exhibited by these four sets of hypophysectomy-responsive DHS mirror the corresponding sex-specific gene response patterns (***Figure 4D***, bottom). Thus, all four DHS sets respond to hypophysectomy in a manner consistent with positively acting regulatory elements linked to sex-biased genes. DHS that are linked to class I sex-biased genes, and whose chromatin closes following hypophysectomy, require pituitary hormone to maintain open chromatin; when pituitary hormones are ablated by hypophysectomy, chromatin closes and their class I sex-biased target genes are repressed. In contrast, DHS that are linked to class II sex-biased genes of the opposite sex bias, and whose chromatin opens following hypophysectomy, are kept in a closed chromatin state by pituitary hormone; when pituitary hormones are ablated, chromatin opens locally at those DHS and their class II sex-biased target genes are de-repressed: expression of female-biased class II genes increases in hypox male liver and expression of male-biased class II genes increases in hypox female liver (***Figure 4—figure supplement 1A***, model).

## A single exogenous GH pulse rapidly reopens chromatin at dynamic DHS in hypox male liver

Next, we investigated whether GH is the pituitary factor whose loss accounts for the widespread chromatin closing seen in hypox male liver. ***Figure 4B*** shows that exogenous GH treatment induces chromatin opening within 30 min at more than 3500 DHS, including 71% of the 1487 dynamic DHS that closed in male liver following hypophysectomy (***Figure 4E***). The fraction of reopening chromatin regions increased to 83% when considering the set of male-biased dynamic DHS that closed following hypophysectomy (***Supplementary file 1A***, column AK). Chromatin reopening was sustained for at least 90 min and then decreased substantially back toward baseline by 240 min (***Figure 4B***; see ***Figure 5A***, plot 1), at which time liver STAT5 signaling has terminated (***Connerney et al., 2017***). Importantly, the mean level of chromatin reopening induced by an exogenous GH pulse at dynamic male-biased DHS was very similar to that of the same DHS set in STAT5-high male liver (***Figure 5A***, plot 1 vs. ***Figure 3A***, plot 5; peak DNase-I activity value 1886 vs. 1848). Thus, the rapid chromatin opening induced by a single exogenous pulse of GH recapitulates the effects of an endogenous GH pulse in opening dynamic male-biased DHS.

Hypophysectomy led to the closure of many fewer static male-biased DHS than dynamic male-biased DHS (***Figure 4C***). Exogenous GH treatment partially reversed chromatin closing at static male-biased DHS within 30 min, with the effect persisting, even after 240 min (***Figure 5A***, plot 2). This

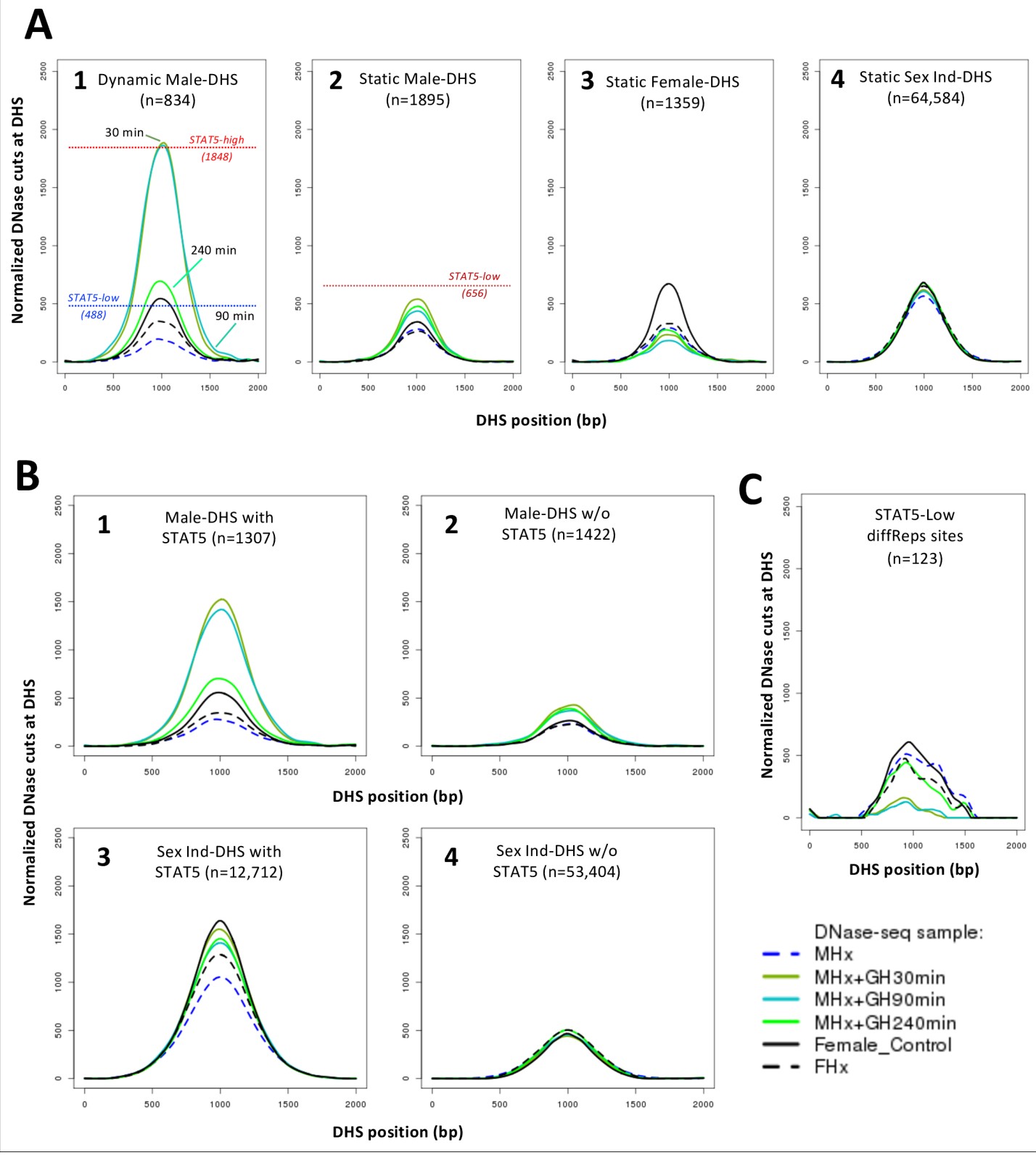

**Figure 5.** DNase-I hypersensitivity site (DHS) activity aggregate plots for hypophysectomy and time course of growth hormone (GH) pulse replacement. (**A**) Normalized DNase-I cut site aggregate plots for each of the indicated sets of static and dynamic DHS showing the effects of hypophysectomy of male (MHx) and female mice (FHx) and of GH pulse treatment (MHx + GH) for 30, 90, and 240 min compared to intact females (c.f., *Figure 3A*). Reference values for normalized DNase-I cut site activity in intact male liver shown in plots 1 and 2 are from *Figure 3* and *Supplementary file 6*. (**B**,

*Figure 5 continued on next page*

*Figure 5 continued*

**C**) Plots as in (**A**) are shown for the indicated subsets of 2729 male-biased DHS (plots 1 and 2) and for the indicated subsets of the set of 66,116 sex-independent DHS (plots 3 and 4), that is, DHS subsets with STAT5 bound (plots 1 and 3) or without STAT5 bound (plots 2 and 4), based on ChIP-seq data for STAT5 binding in intact male mouse liver (*Zhang et al., 2012*), and for the set of 123 STAT5-low sites (see *Figure 2E*).

The online version of this article includes the following figure supplement(s) for figure 5:

**Figure supplement 1.** DNase cut site aggregate plots for the growth hormone (GH) time-course DNase-I hypersensitivity site (DHS) data.

persistence is consistent with static male-biased DHS remaining constitutively open in male liver when liver STAT5 activity dissipates between endogenous plasma GH pulses. Static female-biased DHS also showed a decrease in mean chromatin opening following hypophysectomy, with further decreases in accessibility seen following GH pulse treatment (*Figure 5A*, plot 3). Finally, in an important control, the mean chromatin accessibility at the set of 64,584 static sex-independent DHS was unchanged following hypophysectomy, both with and without or GH pulse treatment (*Figure 5A*, plot 4).

Stratification of the full set of 2729 male-biased DHS by the presence or absence of STAT5 binding, as determined by ChIP-seq, highlighted the STAT5 dependence of the rapid increases in chromatin accessibility stimulated by GH treatment (*Figure 5B*, plot 1 vs. 2). GH-stimulated chromatin reopening at static male-biased DHS was also primarily associated with the STAT5-bound DHS subset (*Figure 5—figure supplement 1*, plot 2A vs. 2B). In contrast, GH induced a much smaller increase in chromatin opening at the STAT5-bound subset of sex-independent DHS (*Figure 5B*, plot 3 vs. 1), where chromatin is already open and largely independent of plasma GH pulses (*Figure 3A*, plot 3). GH decreased chromatin accessibility within 30 min at the 123 genomic sites with lower chromatin accessibility in STAT5-high liver compared to STAT5-low liver (c.f., *Figure 2E*), followed by a return to baseline by 240 min (*Figure 5C*); this response pattern is consistent with the repression of chromatin accessibility at these sites by endogenous GH/STAT5 pulses (*Figure 2E*). Finally, sex-independent DHS that do not bind STAT5 were unresponsive to both hypophysectomy and GH pulse treatment (*Figure 5B*, plot 4).

A subset comprised of 354 dynamic, STAT5-high responsive DHS that close following hypophysectomy did not reopen even 240 min after GH pulse treatment (*Figure 4E*, *Figure 4—figure supplement 2*). These 354 DHS showed significantly lower differential chromatin accessibility between STAT5-high and STAT5-low livers than did the dynamic STAT5-high DHS subset that responded to an exogenous GH pulse (*Figure 4—figure supplement 3A*, set 4 vs. set 5). Conceivably, these 354 dynamic DHS may require multiple GH pulses to reopen or may be co-dependent on other pituitary-regulated hormones ablated by hypophysectomy.

## Distinct sex-biased histone mark patterns at static and dynamic sex-biased DHS

Our initial analyses revealed no major differences between dynamic and static male-biased DHS regarding the distribution of enhancer vs. insulator vs. promoter classifications (*Figure 6—figure supplement 1A*) or their overall chromatin state distributions (*Figure 6—figure supplement 1B*). We therefore examined both classes of male-biased DHS for differences in sex-biased histone marks (*Table 1*). Male-biased enhancer marks (H3K27ac and H3K4me1) were strongly enriched at both dynamic and static male-biased DHS compared to a background set of 64,584 static, sex-independent DHS (*Figure 6A*, *Supplementary file 7B*). In contrast, male-biased H3K36me3 marks, which are characteristic of transcribed regions but have also been shown to inhibit the spread of PRC2-catalyzed H3K27me3 repressive marks (*Yuan et al., 2011*), were enriched at static but not at dynamic male-biased DHS. In addition, static but not dynamic male-biased DHS were significantly depleted of female-biased enhancer marks (H3K27ac and H3K4me1) (*Figure 6B*). This result is consistent with our finding that static male-biased DHS are in a comparatively closed state in female liver (*Figure 3A*, plot 2; model in *Figure 6C*). Static female-biased DHS showed strong enrichments for female-biased enhancer marks (H3K27ac and H3K4me1), female-biased H3K4me3 promoter marks, and female-biased H3K36me3 transcribed region marks (*Figure 6B*), a pattern that is very similar to the enrichments seen at static male-biased DHS (*Figure 6A*).

Examination of two repressive histone marks, H3K27me3 and H3K9me3, revealed their use in a unique way in each sex to enforce sex differences in chromatin states at sex-biased DHS (*Figure 6C*). Male-biased H3K27me3 marks were specifically associated with static female-biased DHS (*Figure 6A*),

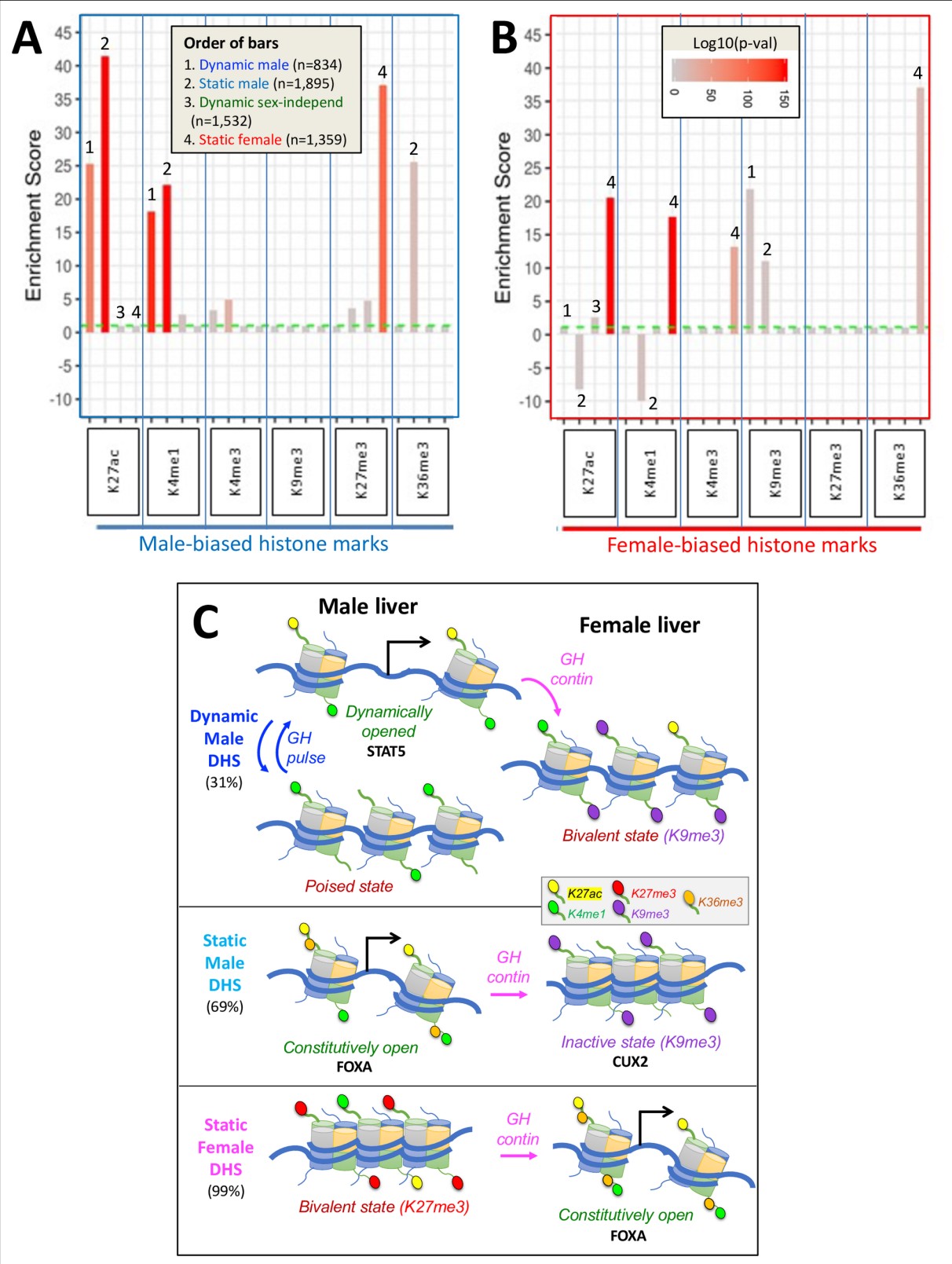

**Figure 6.** Enrichment scores (ES) for sex-biased histone marks at dynamic and static DNase-I hypersensitivity site (DHS) sets. (**A**) Enrichments of male-biased histone marks and (**B**) enrichments of female-biased histone marks. Data is presented as bar graphs showing significant enrichments and significant depletions (negative y-axis values) for the six indicated liver histone marks for each of four DHS sets (see *Figure 2F*). ES are graphed as six sets of four bars each, separated by vertical blue lines and ordered from 1 to 4 (see inset in **A**) and as marked above select bars. The set of

*Figure 6 continued on next page*

*Figure 6 continued*

64,584 static sex-independent DHS was used as the background for the enrichment calculations. Fisher's exact test significance values (log p-values, indicated by bar color; see inset in **B**) are shown for all values that are significant at p<E-03. Values that did not meet this significance threshold are graphed at ES = 1 (horizontal dashed green line); thus, all bars shown, except those graphed at ES = 1.0, represent statistically significant enrichment or depletion. Full details of the number of sites, the source publications used to identify these genomic regions, and corresponding BED files are shown in *Supplementary file 7*. (**C**) Proposed model for chromatin states adopted by dynamic male-biased DHS, static male-biased DHS, and static female-biased DHS in male liver (left) and in female liver (right) in response to the stimulatory and/or repressive actions of plasma GH pulses (in male liver) and persistent GH exposure (in female liver). Histone H3 marks are shown by small colored ovals attached to histone tails (see legend in box). H3K27me3 is specifically used to repress chromatin at female-biased DHS in male liver, and H3K9me3 is specifically used to repress chromatin at both classes of male-biased DHS in female liver. Sex-biased H3K36me3 marks are uniquely associated with static male-biased DHS in male liver and with static female-biased DHS in female liver. They may serve to keep these DHS constitutively open by inhibiting the introduction of H3K27me3 repressive marks (*Yuan et al., 2011*; *Hoetker et al., 2023*) at static female marks in female liver, and perhaps also the introduction of H3K9me3 repressive marks at static male-biased DHS in male liver. Continuous GH infusion in males mimics the female plasma GH pattern and overrides the stimulatory, chromatin opening effects of GH/STAT5 pulses on dynamic male-biased DHS; this, in turn, results in the widespread (95%) closing of dynamic male-biased DHS (*Supplementary file 1E*). DHS with a combination of activating and repressive histone marks in one but not both sexes (i.e., sex-dependent bivalent character) are indicated. The degree of chromatin accessibility is indicated by the relative distance between nucleosomes. Black arrows indicate DHS stimulation of gene transcription upon interaction of these enhancer DHS with a nearby or distal gene promoter. We speculate, but have not tested experimentally, that GH pulses induce an increase in activating histone marks at dynamic male-biased DHS, as indicated by the increase in H3K27ac marks shown here when the dynamic male biased DHS are opened.

The online version of this article includes the following figure supplement(s) for figure 6:

**Figure supplement 1.** Chromatin state analysis of static and dynamic male-biased DNase-I hypersensitivity site (DHS).

consistent with our prior work showing deposition of these marks by the Ezh1/Ezh2 enzymatic component of PRC2 as a specific mechanism to repress many female-biased genes in male liver (*Lau-Corona et al., 2020*). In contrast, female-biased H3K9me3 repressive marks were significantly enriched at both dynamic and static male-biased DHS (*Figure 6B*); however, we did not find any corresponding enrichment of male-biased H3K9me3 marks at female-biased DHS. This novel finding suggests that H3K9me3 marks are specifically used to repress male-biased genes in female liver, contrasting with the specific use of H3K27me3 marks to repress female-biased genes in male liver.

## Chromatin state analysis uncovers bivalent-like states at closed sex-biased DHS

We sought to delineate chromatin state differences at sex-biased DHS in each sex by computing the enrichment (or depletion) of each of 14 distinct chromatin states for each DHS set. These 14 chromatin states are defined by a panel of activating and repressive histone-H3 marks and by the presence of open chromatin (*Figure 1C*) and were determined separately for male and female mouse liver (*Sugathan and Waxman, 2013*). Enrichments were calculated using a genome-wide background set comprised of all 64,584 static sex-independent DHS (*Figure 7*). In male liver, enhancer state E6 (characterized by the presence of DHS, H3K27ac and H3K4me1; *Figure 1C*) was moderately enriched compared to the background DHS set at both dynamic and static male-biased DHS, and at dynamic sex-independent DHS. Promoter states E7 and E8 were significantly depleted, consistent with the paucity of promoter states in the full set of male-biased DHS (*Figure 1A*). Dynamic but not static male-biased DHS showed strong depletion in male liver of the inactive chromatin states E1 (H3K27me3 marks) and E2 (major marks not identified) and of state E14 (H3K36me3 marks) (*Figure 7A*). Finally, we observed strong enrichment of the inactive state E2 at both dynamic and static male-biased DHS in female liver (*Figure 7B*).

Dynamic male-biased DHS were enriched for the enhancer state E9 in female liver (ES = 2.42, p=1.6E-69; *Figure 7B*) but not male liver (*Figure 7A*). The emission parameters of state E9 are very similar to those of state E6, namely, it shows a high frequency of the activating chromatin marks H3K27ac and H3K4me1 but lacks the open chromatin (DHS) feature characteristic of state E6 (*Figure 1C*). Thus, in female liver, dynamic male-biased DHS contain histone marks that typify an active enhancer but are inactive due to the comparatively closed state of their chromatin. Indeed, the extent of chromatin opening at these DHS in female liver is equivalent to the level of chromatin opening seen at the same set of sites in male liver between GH/STAT5 activity pulses (*Figure 3A*, plot 5; model in *Figure 6C*). Dynamic male-biased DHS were also enriched for chromatin state E10 in

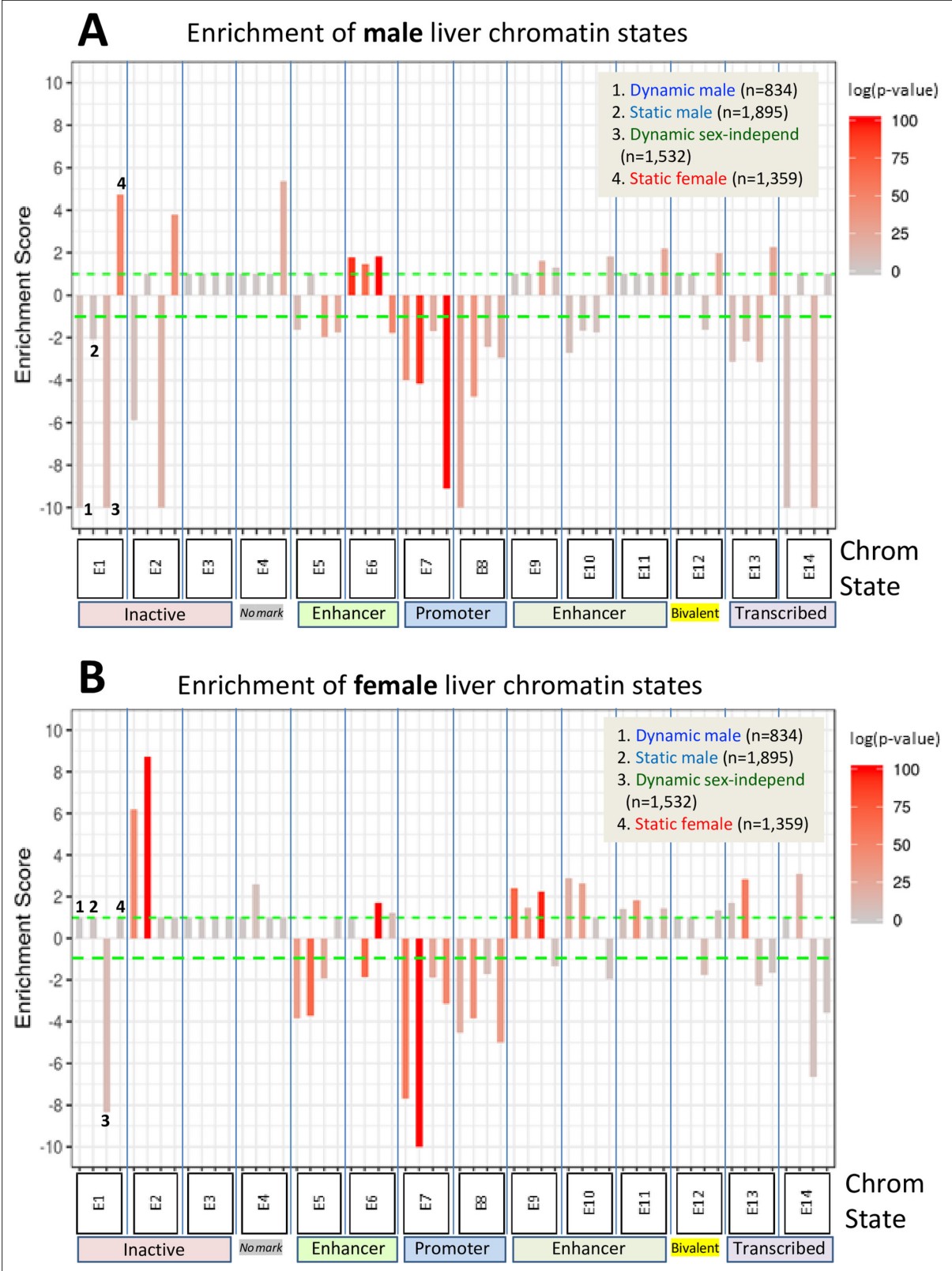

**Figure 7.** Enrichment scores for chromatin states at sex-biased and sex-independent dynamic and static DNase-I hypersensitivity sites (DHS). Shown are the enrichments of male liver chromatin states (**A**) and of female liver chromatin states (**B**) at each of the four indicated DHS sets. Data are presented as described in *Figure 6*, with the set of 64,584 static sex-independent DHS used as background for the enrichment calculations. As many of the background set of DHS are active regulatory regions replete with enhancer marks, it is to be expected that the dynamic and static sex-biased DHS sets

*Figure 7 continued on next page*

*Figure 7 continued*

would show low, albeit significant enrichments for enhancer states E5, E6, and E9–E11. Full details of these analyses, including DHS chromatin states and source publications used to identify these genomic regions, are provided in *Supplementary file 7*.

female liver (ES = 2.89, p=1.9E-18) but not male liver. State E10 shows a high frequency of H3K27ac marks, but not H3K4me1 marks, and lacks DHS. This pattern of chromatin state E9 and E10 enrichment at dynamic male-biased DHS in female liver, combined with the strong enrichment of H3K9me3 repressive marks (*Figure 6B*), indicates that the genomic regions encompassing dynamic male-biased DHS are in a bivalent-like chromatin state in female liver (*Figure 6C*). Such a bivalent state could facilitate chromatin opening under certain pathological conditions, for example, in response to foreign chemicals or biological stress.

Static female-biased DHS showed little or no enrichment of specific chromatin states in female liver compared to the genome-wide background set of static sex-independent DHS (*Figure 7B*). This supports a model whereby the female bias in accessibility at these sites is largely due to their active suppression in male liver (*Figure 6C*). Indeed, the genomic regions encompassing static female-biased DHS were strongly enriched in male liver for inactive chromatin states E1 (repressive mark H3K27me3), E2, and E4 (*Figure 7A*). Static female-biased DHS showed moderate enrichment in male liver for enhancer states E10 and E11, which are respectively characterized by the active enhancer marks H3K27ac and H3K4me1 but devoid of DHS. This is analogous to our finding, above, that male-biased DHS are enriched for chromatin states replete with active histone marks but deficient in DHS in female liver. Static female-biased DHS also showed moderate enrichment in male liver for bivalent state E12, which is characterized by a mixture of activating histone marks and the presence of repressive H3K27me3 marks, consistent with the strong enrichment of the latter histone mark at static female-biased DHS seen in *Figure 6A*, above, and the overall conclusion that at least a subset of static female-biased DHS is in a bivalent-like state in male liver. This pattern is analogous to the bivalent state adopted by dynamic male-biased DHS in female liver, except for the use of distinct marks to effect sex-specific repression in each sex, namely: H3K9me3 in female liver and H3K27me3 in male liver (*Figure 6C*). Finally, promoter-like states E7 and E8 were strongly depleted from static female-biased DHS (*Figure 7*), consistent with female-biased DHS largely being gene distal sex-biased enhancers.

## Distinct gene targets and enriched biological processes of dynamic and static sex-biased DHS

Dynamic and static male-biased DHS both showed strong enrichment for mapping to male-biased gene targets (8-fold and 11.7-fold enrichments, respectively), as did static female-biased DHS for female-biased gene targets (12.5-fold enrichment) when DHS were mapped to the single nearest transcription start site in the same TAD (*Supplementary file 1B*). Further, mapping DHS to putative target genes using GREAT, which typically maps each DHS to two genes, revealed many examples where male-biased genes were predicted to be regulated by multiple male-biased DHS. In some cases, all of the associated male-biased DHS were static male-biased DHS (e.g., *Cyp4a12a*, which mapped to seven static male-biased DHS), while in other cases they were a mixture of dynamic and static male-biased DHS (e.g., *Cyp7b1*, with seven dynamic and eight static male-biased DHS) (*Supplementary file 9A*). Top female-biased genes enriched for nearby static female-biased DHS included *Cux2* and three *Cyp3a* genes, each of which mapped to 10–15 female-biased DHS (*Supplementary file 10C*). Sex-independent genes that are well-established direct targets of STAT5 (*Rotwein, 2020*) include *Igf1*, a target of 12 dynamic sex-independent DHS and 3 dynamic male-biased DHS, and *Socs2*, a target of 8 dynamic sex-independent DHS (*Supplementary file 10A and D*). Other analyses revealed that all three sets of sex-biased DHS were significantly enriched for mapping to hypophysectomy class I-responsive sex-biased genes compared to hypophysectomy class II-responsive sex-biased genes (*Supplementary file 1B*). This finding supports the conclusion that these sex-biased DHS sets are enriched for positively acting enhancers that close following hypophysectomy, which leads to decreased expression of their class I (i.e., hypophysectomy repressed) sex-biased gene targets.

The gene targets of each sex-biased DHS set showed significant enrichment for distinct but partially overlapping Gene Ontology (GO) Biological Processes, as determined by GREAT analysis. Top enriched GO terms common to both dynamic and static male-biased DHS included lipid metabolic process and steroid metabolic process, consistent with the major role that GH plays in

the male-prevalence of fatty liver development and liver metabolic disease (*Dichtel et al., 2022*; *Kaltenecker et al., 2019*; *Oxley et al., 2023*). Unique enriched terms for dynamic male-biased DHS included cell adhesion, gland development, and hepatico-biliary development, whereas gene targets of static male-biased DHS were uniquely enriched for cellular response to glucocorticoid stimulus and EGF receptor signaling, among others (*Figure 2—figure supplement 2*, *Supplementary file 10*, bottom). Static female-biased DHS gene targets were uniquely enriched for long-chain fatty acid metabolic pathway and related terms. Finally, the set of dynamic sex-independent DHS mapped to gene targets enriched for terms related to glucose transport and metabolism, IGF receptor signaling, and fibroblast proliferation (*Figure 2—figure supplement 2*). This finding is consistent with the widespread metabolic effects that GH has in both sexes, including complex effects on glucose uptake and glucose oxidation, and on hepatic gluconeogenesis and glycogenolysis (*Kim and Park, 2017*).

## Discussion

Sex differences in chromatin accessibility are a central epigenetic feature that enables regulatory proteins, such as the GH-activated transcription factor STAT5, to bind chromatin and regulate hepatocyte gene transcription in a sex-specific manner. However, the underlying mechanisms controlling sex differences in liver chromatin accessibility, which occur at more than 4000 distinct sites across the mouse genome, are poorly understood. GH, working through its sex-dependent temporal patterns of secretion by the pituitary gland, is the major hormonal factor controlling STAT5-dependent sex differences in liver gene transcription, but little is known about the potential of GH to directly regulate sex differences in the epigenetic landscape required for sex-dependent transcriptional outputs. To address this question, we elucidated the impact of male plasma GH pulses on global patterns of liver chromatin accessibility by analyzing livers from a population of 18 individual male mice euthanized at either a peak or a trough of plasma GH pulse-stimulated hepatic STAT5 DNA-binding activity. Our findings establish that the naturally occurring, endogenous plasma GH pulses characteristic of males induce dynamic cycles of chromatin opening and closing at several thousand DHS in male mouse liver and that these events comprise one of two major mechanisms regulating the male bias in liver chromatin accessibility. Analysis of sex-dependent transcription factor binding patterns, histone marks, and chromatin states elucidated key features distinguishing this dynamic mechanism of male-biased enhancer activation from that of static, GH pulse-unresponsive male-biased DHS and from female-biased DHS (see model, *Figure 6C*). GH thus acts at three distinct steps to regulate sex-dependent hepatocyte gene expression, all three involving the GH-activated transcription factor STAT5 (*Figure 8*): GH pulse-induced chromatin opening at dynamic male-biased DHS driven by pulsatile GH activation of STAT5, as discussed further below; direct transcriptional activation of sex-biased genes by GH-activated STAT5; and GH/STAT5-dependent transcriptional regulation of downstream repressors, such as the female-specific CUX2, which binds to male-biased enhancers in female liver to reinforce sex differences in gene transcription.

### Endogenous GH/STAT5 pulses induce repeated chromatin opening and closing at dynamic male-biased DHS

GH pulse-induced chromatin opening in male mouse liver is shown to be directly controlled by the endogenous male rhythm of plasma GH stimulation of hepatocytes, the major liver cell type contributing to sex-biased liver gene expression (*Goldfarb et al., 2022*). The GH pulse-regulated chromatin regions identified here responded to changes in plasma GH levels in a rapid and dynamic manner, with extensive GH-induced chromatin opening occurring within 30 min, as seen when hypophysectomized male mice were given a physiological replacement dose of GH pulse by intraperitoneal injection (*Figure 4A*). This time course is consistent with the rapid activation of GH receptor signaling to STAT5, which occurs within 5 min in cell culture (*Gebert et al., 1997*) and within 15 min in vivo in a hypophysectomized rat model (*Waxman et al., 1995*). Importantly, 85% of the dynamic male-biased DHS identified here (710 of 834 dynamic male-biased DHS) were bound by STAT5, which appears to be a key driver of these GH pulse-induced chromatin opening events. Chromatin closing followed the termination of nuclear STAT5 signaling, which is complete within 4 hr (*Connerney et al., 2017*) and occurs in sufficient time to reset the GH receptor-JAK2 signaling complex and resensitize hepatocytes before the next plasma GH pulse (*Gebert et al., 1997*; *Figure 2A*). STAT5 binding and chromatin

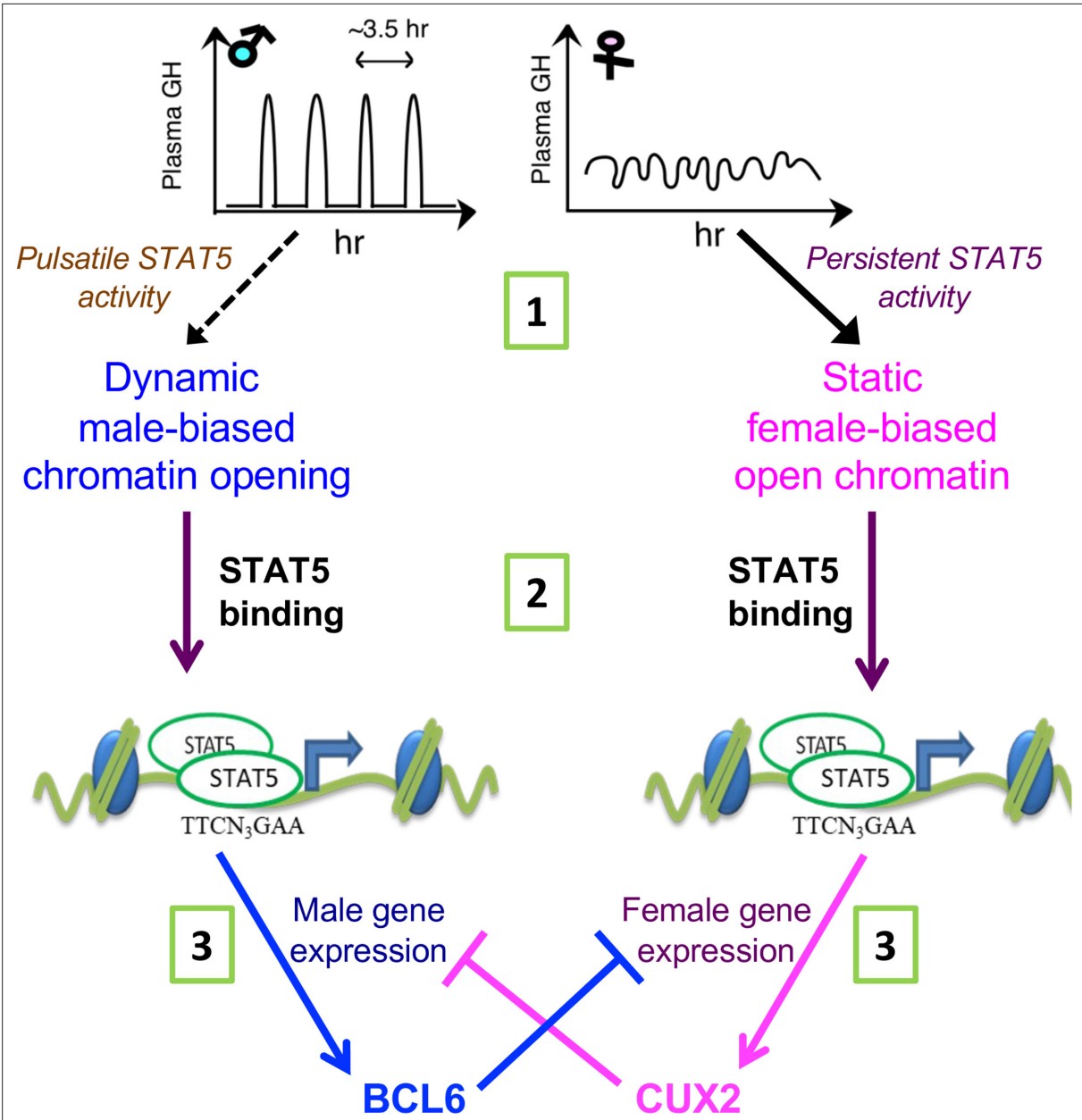

**Figure 8.** STAT5 regulates sex-dependent hepatocyte gene expression at three distinct steps. (1) Sex-biased chromatin opening: growth hormone (GH) pulse-induced chromatin opening at dynamic male-biased DNase-I hypersensitivity sites (DHS) is driven by pulsatile GH activation of STAT5 in male liver, whereas persistent activation of STAT5 in female liver is associated with static female-biased chromatin opening. (2) Sex-biased transcriptional activation: sex differences in open chromatin regions and their accessibility enable GH-activated STAT5, and other transcription factors, to bind chromatin in a sex-biased manner and induce the transcriptional activation of sex-biased genes. (3) Sex-based transcriptional repression: the sex-biased regulatory genes regulated in step (2) include the GH/STAT5-dependent repressor proteins BCL6 (male-biased) and CUX2 (female-specific), which reinforce sex differences in transcription by preferentially suppressing the expression of female-biased and male-biased genes, respectively, as indicated.

opening at dynamic male-biased DHS were both significantly lower in livers of female mice, where plasma GH levels and liver STAT5 activity are persistent (near-continuous) yet ineffective at maintaining an open chromatin state. Indeed, the extent of chromatin opening at these male-biased DHS in female liver is very similar to the level in livers from male mice euthanized when liver STAT5 activity is low, a time inferred to be between plasma GH pulses (STAT5-low male livers; *Figure 3A*, panel 5). Pulsatile chromatin opening stimulated by endogenous plasma GH pulses is thus a unique mechanism

for establishing and maintaining male-biased chromatin accessibility and transcription factor binding at 834 male-biased DHS, corresponding to 31% of all male-biased DHS in the liver.

GH pulse-responsive DHS were discovered by comparing global chromatin accessibility patterns in a set of 8 livers collected from mice euthanized at a peak of GH-activated STAT5 DNA-binding activity (STAT5-high livers) to those of 10 other livers from mice euthanized between pulses of STAT5 DNA-binding activity, that is, when liver STAT5 activity is very low or undetectable (STAT5-low livers). Three other male livers gave DNase-seq profiles inconsistent with their EMSA-determined STAT5 DNA-binding activity. These outlier livers may have come from male mice euthanized just after the onset of a STAT5 pulse (*Tannenbaum et al., 2001*), when more time is needed to induce chromatin opening (two outliers), or shortly after STAT5 is deactivated by tyrosine dephosphorylation (*Able et al., 2017*) but prior to the reversal of chromatin opening (one outlier). Importantly, the top 200 genomic regions showing differential chromatin opening between STAT5-high and STAT5-low livers also separated a group of nine C57Bl/6 male mouse liver DNase-seq samples from the ENCODE consortium into two distinct classes, corresponding to the accessibility patterns of STAT5-high-activity livers (n = 4) and STAT5-low-activity livers (n = 5), respectively. Thus, the plasma GH-induced signaling pathways and downstream epigenetic events leading to dynamic chromatin opening and closing at these specific genomic regions are robust across studies and mouse strains.

STAT5-high male livers showed the highest levels of chromatin accessibility among the male-biased DHS that bind STAT5. Similarly, a higher level of chromatin opening was seen at sex-independent DHS that bind STAT5 compared to those that do not bind STAT5 (*Figure 3A*, panel 3 vs. 4). There is thus a close linkage between STAT5 binding and the extent of chromatin opening. This finding is consistent with the proposal that STAT5 acts as a pioneer factor to enable chromatin opening, as was reported for STAT5 action at the *Il9* gene locus in Th9 T-cells (*Fu et al., 2020*). It should be noted, however, that pulsatile chromatin opening also occurred at a small subset (15%) of dynamic male-biased DHS that do not bind STAT5, suggesting that other GH receptor signaling pathways (*Frank, 2020*) play a role in male-biased chromatin opening at those sites. It is also apparent that pulsatile STAT5 activation and pulsatile STAT5 DNA binding alone are not sufficient to ensure pulsatile chromatin opening insofar as many sex-independent DHS do bind STAT5, yet are constitutively open in both male and female liver (*Figure 3—figure supplement 1C*, plot 5A). Finally, plasma GH pulse stimulation decreased chromatin accessibility at a small number of liver DHS, most of which were sex-independent. The mechanism for this GH-stimulated decrease in accessibility and its physiological significance are unknown.

## Sex-dependent GH regulation of static male-biased DHS

We identified a second, less well-defined mechanism that controls the male bias in chromatin accessibility at a distinct DHS set, comprised of 1895 static male-biased DHS. These DHS, which represent 69% of all male-biased DHS, mapped to a set of target genes and enriched biological processes distinct from but overlapping with those of the dynamic male-biased DHS set. These static male-biased DHS are constitutively open in male liver, with chromatin accessibility largely unchanged across the peaks and valleys of GH-induced liver STAT5 activity. Nevertheless, GH regulates the sex bias of these DHS, as evidenced by their extensive closure in livers of male mice given a continuous infusion of GH (*Supplementary file 1E*), which mimics the female plasma GH pattern and substantially feminizes liver gene expression (*Lau-Corona et al., 2017*; *Lau-Corona et al., 2022*). Thus, in contrast to dynamic male-biased DHS, the sex-biased chromatin accessibility of static male-biased DHS is primarily due to their closed chromatin state in the persistent GH signaling environment of female liver (*Figure 6C*). Static male-biased DHS are relatively deficient in STAT5 binding compared to dynamic male-biased DHS, but showed significant enrichment for binding the female-specific repressor protein CUX2 (*Conforto et al., 2012*), which we infer contributes to their closure in both female liver and in continuous GH-infused male liver (where CUX2 is induced to female-like levels; *Lau-Corona et al., 2017*; *Laz et al., 2007*) through its established transcriptional repressor activity (*Gingras et al., 2005*). Finally, static male-biased DHS showed strong enrichment of male-biased binding sites for the pioneer factors FOXA1 and FOXA2, which may help maintain their constitutively open chromatin state in male liver.

## Novel insights into the underlying mechanisms from chromatin state analysis

Both dynamic and static male-biased DHS are largely (~95%) promoter-distal enhancer DHS, as indicated by their flanking histone modifications. This supports our prior finding that sex-dependent intra-TAD DNA looping mechanisms are common in mouse liver (*Matthews and Waxman, 2020*) and indicates such looping likely bring both sets of male-biased DHS into closer proximity with their sex-biased promoter targets. Dynamic male-biased DHS were enriched for the active enhancer state E6 in male liver but were enriched for enhancer states E9 and E10 in female liver. Enhancer states E9 and E10 are characterized by a high frequency of same activating chromatin marks as chromatin state E6, namely H3K27ac and H3K4me1 (E9) or H3K27ac alone (E10), but unlike E6 they are both deficient in open chromatin (DHS) (*Figure 1C*). Thus, in female liver, dynamic male-biased DHS regions contain active histone marks but are in a comparatively closed chromatin state. This may in part be due to their enrichment for female-biased H3K9me3 (repressive) histone marks, which gives them bivalent character (*Figure 6C*). A bivalent chromatin state may protect dynamic male-biased DHS from irreversible silencing (*Kumar et al., 2021*) and confer the potential for chromatin opening leading to activation of their latent enhancer activity in female liver, for example, in response to chemical exposure or biological stress (*Lodato et al., 2018*). In contrast, static male-biased DHS are depleted of female-biased enhancer marks (H3K27ac and H3K4me1) and are in an inactive chromatin state in female liver. Notably, the bivalent female chromatin state of dynamic male-biased DHS is distinct from the poised state these genomic regions apparently adopt in male liver between plasma GH pulses (i.e., in STAT5-low male liver) (*Figure 6C*) and in hypophysectomized male liver, where a single GH pulse is all that is required to induce rapid chromatin opening, even after several weeks of pituitary hormone ablation.

The set of female-biased DHS identified here is almost entirely (99%) static, that is, GH pulse-unresponsive, and showed strong enrichment for three female-biased activating chromatin marks, H3K27ac, H3K4me1, and H3K4me3. These DHS also showed strong enrichment for female-biased FOXA2 binding, which likely contributes to their high chromatin accessibility in female compared to male liver (*Figure 6C*). The static female-biased DHS also showed strong enrichment for the repressive histone mark H3K27me3 in male liver, which is expected to contribute directly to their closed, inactive male liver chromatin state. Moderate enrichments for several chromatin states characterized by the presence of activating chromatin marks but lacking in DHS (states E10, E11, and E12) were also observed. This indicates heterogeneity within the set of static female-biased DHS, some of which have bivalent chromatin character in male liver. Evidence for heterogeneity of these female-biased DHS also comes from their varied time-dependent loss of H3K27me3 marks and from their differential time courses for chromatin opening when male mice are given GH as a continuous infusion (*Lau-Corona et al., 2017*). Heterogeneity was also seen in the extent of female-biased target gene de-repression when H3K27me3 marks are lost from livers of male mice deficient in the H3K27-trimethylase enzyme complex PRC2 (*Lau-Corona et al., 2020*). Further study is needed to elucidate the mechanistic basis for these time-dependent epigenetic responses and their associated sex-biased gene expression changes (*Lau-Corona et al., 2017*).

Novel insight into the fundamental underlying epigenetic mechanisms of sex-biased gene regulation comes from our discovery that distinct repressive histone marks, H3K27me3 and H3K9me3, are used in a unique way in each sex to enforce sex differences in chromatin states at sex-biased DHS. Whereas male-biased H3K27me3 repressive marks are highly enriched at and are specifically associated with static female-biased DHS, in agreement with our prior work (*Sugathan and Waxman, 2013*; *Lau-Corona et al., 2020*), we show here that female-biased H3K9me3 repressive marks are specifically enriched at both dynamic and static male-biased DHS, without a corresponding enrichment of male-biased H3K9me3 marks at static female-biased DHS. This unexpected finding supports the proposal that sex-specific gene repression is mediated by two distinct mechanisms: H3K9me3 marks are specifically used to repress certain male-biased regulatory sites in female liver, and H3K27me3 marks are specifically used to repress female-biased regulatory sites in male liver.

We also discovered that sex-biased H3K36me3 marks are a unique distinguishing feature of static sex-biased DHS, with male-biased H3K36me3 marks being highly enriched at static male-biased DHS but not at dynamic male-biased DHS, and female-biased H3K36me3 marks highly enriched at static female-biased DHS (*Figure 6C*). H3K36me3 marks are classically associated with the demarcation of

actively transcribed genes (*Li et al., 2019*) but are also used to maintain cell type identity by inhibiting the spread of H3K27me3 repressive marks at cell type-specific enhancers (*Yuan et al., 2011*; *Hoetker et al., 2023*). The enrichment of H3K36me3 marks at static male-biased DHS described here could thus be an important mechanism to maintain sex-dependent hepatocyte identity by keeping static male-biased enhancers constitutively open and free of H3K27me3 repressive marks in male liver, and correspondingly for H3K36me3 marks enriched at static female-biased DHS in female liver. Further study is needed to elucidate the mechanisms whereby these and the other sex-specific histone marks discussed above are deposited on chromatin in a sex-dependent and site-specific manner and the roles that GH plays in regulating these epigenetic events.

## Regulatory elements associated with class I and II sex-biased genes

DNase-seq profiling identified more than 5000 liver DHS that open or close following hypophysectomy. Strikingly, a single physiological replacement dose of GH restored, within 30 min, liver STAT5 activity and chromatin accessibility at 83% of the dynamic male-biased DHS that closed following hypophysectomy. These chromatin sites are maintained in a primed (poised) state in hypox male liver over a period of several weeks, despite the prolonged deficiency of GH and other pituitary hormones, including gonadal steroids, and the dysregulation of several thousand liver-expressed genes (*Wauthier et al., 2010*; *Connerney et al., 2017*). DHS that close in male liver following hypophysectomy were enriched for proximity to class I male-specific genes, which are downregulated in hypox male liver, and DHS that close in hypox female liver showed the strongest enrichment for class I female-specific gene targets, which are downregulated in hypox female liver (*Figure 4—figure supplement 1A*, model). Furthermore, DHS that open in male liver following hypophysectomy were enriched for proximity to class II female-specific genes and DHS that open in female liver were enriched for proximity to class II male-specific genes (*Figure 4D*). These enrichments are consistent with the definition of class II sex-biased genes, that is, genes that are upregulated by hypophysectomy in the sex where they show lower expression in intact mice. Together, these findings lend strong support for the functional role of each class of sex-biased and pituitary hormone-dependent DHS as regulatory elements with intrinsic, positively acting enhancer potential, keeping gene expression on in intact liver by maintaining open chromatin, for example, at male-biased DHS repressed in hypox male liver in the case of class I male-biased genes, or keeping gene expression off by maintaining a closed chromatin state, for example, at female-biased DHS induced in hypox male liver in the case of class II female-biased genes. Further study will be required to elucidate the regulatory factors and molecular mechanisms governing these gene regulatory circuits, in particular those controlling the de-repression of class II sex-biased genes following pituitary hormone ablation.

## Pituitary GH secretory patterns vs. gonadal steroids as regulators of sex-biased liver chromatin accessibility and gene expression

Testosterone has a well-established role in early postnatal programming of the hypothalamic control of pituitary GH secretory patterns beginning at puberty and lasting through adulthood (*Chowen et al., 1996*; *Ramirez et al., 2010*; *Waxman et al., 1985*). While it is also possible that testosterone, as well as estrogens, could also regulate sex differences in hepatocytes directly at the epigenetic or transcriptional level, our findings support the proposal that plasma GH patterns, and not gonadal steroids, dominate the epigenetic control of liver sex differences. First, the ability of a single exogenous plasma GH pulse to rapidly reopen dynamic male-biased DHS closed by hypophysectomy – in the face of ongoing ablation of pituitary stimulated gonadal steroid production and secretion – implicates GH signaling *per se* in the direct regulation of chromatin accessibility for this class of male-biased DHS. Second, the sex biased accessibility of static male-biased DHS is also regulated by the pattern of plasma GH stimulation, as evidenced by the widespread closure of those DHS in male liver following continuous GH infusion (*Supplementary file 1E*). On the other hand, it is important to note that hepatocyte-specific knockout of androgen receptor (AR) does, in fact, dysregulate ~15% of sex-biased genes in the liver, albeit with a much lower effect size than global AR knockout (*Xiong et al., 2023*), which may be due to disruption of the somatotropic axis and circulating GH secretory profiles in the case of global AR loss. Conceivably, AR could regulate these sex-biased genes via a direct binding mechanism, by acting alone, or perhaps in concert with GH-activated STAT5, to keep chromatin open constitutively at a subset of static male-biased DHS, of which 32% undergo at least

partial closure in male liver following hypophysectomy (*Figure 4C*). Supporting this possibility, AR binding sites were recently reported to be enriched at male-biased genes in mouse liver (*Rodríguez-Montes et al., 2023*). In contrast, estrogen receptor (ERα) likely plays only a minor role in regulating sex-biased liver DHS enhancers, given the lack of major effect of hepatocyte-specific ERα knockout on sex-biased liver gene expression (*Nikkanen et al., 2022*) and our finding that only 12% of static female-biased DHS close in female liver following hypophysectomy, despite the decrease in circulating estradiol levels (*Wang and Greenwald, 1993*).

## Methods

### Animal treatments and EMSA analysis

All mouse work was carried out in accordance with ARRIVE Essential guidelines 2.0 (*Percie du Sert et al., 2020*) for study design, sample size, randomization, experimental animals and procedures, and statistical methods, and with approval of the Boston University Institutional Animal Care and Use Committee. Male and female CD1 mice (ICR strain, strain code # 022), 7-8 weeks old and purchased from Charles River Laboratories (Wilmington, MA), were kept on a 12 hr light/dark cycle with food and water without restriction. Livers were collected from individual untreated mice between 8 and 10 wk of age. Where indicated, male and female mice were hypophysectomized (hypox) by the supplier at ~7–8 wk of age. Completeness of hypophysectomy was verified by the absence of weight gain over a 2–3-week period and by the lack of Mup protein in urine samples (SDS-PAGE analysis) (*Connerney et al., 2017*). Hypox male mice were treated with recombinant rat GH (purchased from Prof. Arieh Gertler, Protein Laboratories Rehovot Ltd, Rehovot, Israel), given as a single intraperitoneal injection at 125 ng of GH per gram body weight (*Wauthier et al., 2010*), or vehicle (control), and were euthanized by cervical dislocation 30, 90, or 240 min later (*Connerney et al., 2017*). This dose is 12 times lower than the supraphysiological dose of GH widely used to study reactivation of hepatic *Igf1* following hypophysectomy (*Alzhanov et al., 2015*). A protein extract prepared from a small piece of each liver to be used for DNase-seq analysis (see below) was assayed for liver STAT5 DNA-binding activity by EMSA, as described (*Connerney et al., 2017*). Results of this assay allowed us to classify each individual male mouse as STAT5-high activity (n = 10) or STAT5-low activity (n = 11) at the time of euthanasia and liver collection. EMSA was also used to verify the ablation of liver STAT5 activity following hypophysectomy and the effectiveness of exogenous GH administration at restoring endogenous STAT5-high liver levels of STAT5 EMSA activity within 30 min.

### DNase-seq analysis, Illumina sequencing, and data processing

Livers used for these analyses were obtained from intact male and female mice, from hypox male and hypox female mice, and from hypox male mice given a single injection of GH and euthanized 30, 90, or 240 min later (n = 6–12 livers per group). Nuclei were purified from individual fresh livers as described (*Ling et al., 2010*) and stored at –80°C. Nuclei were subsequently treated with DNase I to release DNA fragments from hypersensitive genomic regions to identify liver DHS (*Connerney et al., 2017*). The DNA fragments released from each liver were combined (three livers per pool) to give n = 2 to n = 4 independent pooled samples, which were used to prepare DNase-seq libraries for each mouse group (*Supplementary file 4*). In the case of STAT5-high and STAT5-low liver DHS analysis, however, DNase-seq libraries were prepared directly from DNase-I digested fragments released from each individual liver without pooling, as described below. DNase-seq libraries were prepared using the NEBNext Ultra Library Prep Kit for Illumina (New England Biolabs). Illumina sequencing was performed on an Illumina HiSeq instrument to a depth of 12–33 million mapped 40–50 bp single-end sequence reads per sample. Detailed sequencing statistics are shown in *Supplementary file 4*.

Sequencing data was analyzed using a custom in-house DNase-seq pipeline (*Lodato et al., 2018*). Briefly, the pipeline processes raw FASTQ files and outputs various control metrics, including FASTQC reports, confirmation of read length, verification of the absence of read strand bias, and identification of contaminating adaptor sequences using Trim Galore, *RRID*:SCR_011847 (https://github.com/FelixKrueger/TrimGalore; *Krueger, 2023*). Reads were mapped to mouse genome mm9 using Bowtie2 (v2.2.6) (*Langmead and Salzberg, 2012*). Regions of DNase hypersensitivity (DHS) were discovered as peaks identified by MACS2 (v2.1.0.20150731) (*Feng et al., 2012*) using the option (-nomodel –shift –100 –extsize 200) to inhibit read shifting, and the option (-keep-dup) to retain all

reads that contribute to the peak signal. Peaks were discovered for each individual DNase-seq library then filtered to remove peaks that overlap ENCODE blacklisted regions (*Landt et al., 2012*), as well as peaks comprised of five or more identical reads that do not overlap any other read ('straight peaks'). Nine additional individual male mouse liver DNase-seq datasets generated by the ENCODE consortium (FASTQ files downloaded here; GEO accession: GSM1014195, liver replicates #5 to #13) were analyzed using the same DNase-seq pipeline (cell line, liver; strain, C57BL/6; age, adult 8 wk; sex, male). These DNase-seq datasets were single-end sequencing reads, 36 bp in length, with a sequencing depth of 19–37 million total mapped reads per sample.

## Liver DHS classification and DHS target gene assignment

A set of 72,862 mouse liver DHS regions previously identified by DNase-seq analysis of male and female mouse liver (*Ling et al., 2010*) was filtered to remove 515 DHS regions that overlapped ENCODE blacklisted regions. A total of 2136 DHS regions that could not be classified according to a five-class DHS model defined previously (*Matthews and Waxman, 2018*) were also removed to obtain a standard reference set comprised of 70,211 mouse liver DHS. Each DHS was designated as a promoter, weak promoter, enhancer, weak enhancer, or insulator DHS based on the five-class model, which is primarily based on ChIP-seq signals for the H3 histone marks H3K4me1and H3K4me3, combined with CTCF ChIP-seq binding in adult male mouse liver (*Matthews and Waxman, 2018*). Weak enhancers were identified by their low levels of H3K27ac ChIP-seq signal compared to DHS classified as enhancers, combined with a distance from RefSeq gene transcription start sites inconsistent with a weak promoter designation (*Matthews and Waxman, 2018*). Further, individual DHS were designated male-biased (n = 2729), female-biased (n = 1366), or sex-independent (n = 66,116) based on significant differences in chromatin accessibility between male and female mouse liver (*Ling et al., 2010*). Each DHS was assigned a single putative gene target (RefSeq gene or multi-exonic lncRNA gene [*Melia and Waxman, 2019*]) corresponding to the closest transcription start site within the same TAD (*Matthews and Waxman, 2018*), except as noted. *Supplementary file 1A* lists DHS target genes, their overlap with liver STAT5 binding sites determined by ChIP-seq (*Zhang et al., 2012*), their DNase-seq activity (chromatin accessibility) in male and female mouse liver, and their responses to hypophysectomy and to hypophysectomy + GH treatment. For some analyses, including Gene Ontology (GO) enrichments, GREAT (v.4.0.4) (http://great.stanford.edu/public/html/index.php; *McLean et al., 2010*; *Tanigawa et al., 2022*) was used to identify target genes using default settings for the Basal + extension gene association rule, which typically assigns two genes to each DHS. GREAT output is provided in *Supplementary files 9 and 10*.

## Differential DHS between STAT5-high- and STAT5-low-activity livers

Nuclei purified from each individual STAT5-high-activity (n = 10) and STAT5 low-activity mouse liver (n = 11), classified based on EMSA as described above, were analyzed by DNase-seq to discover genomic regions (i.e., DHS) that responded dynamically to endogenous plasma GH pulses and the associated changes in liver STAT5 DNA-binding activity, as follows. diffReps analysis (*Shen et al., 2013*) was used to discover genomic sites that were more open or were more closed (DNase-seq normalized intensity |fold-change|> 2 and FDR < 0.05 [Benjamini–Hochberg adjusted p-value]) for the set of EMSA-identified STAT5-high-activity livers compared to the set of EMSA-identified STAT5-low-activity livers. The diffReps nucleosome option (200 bp window size) was used and the option (-frag) was set to zero for all comparisons. The differential sites that diffReps identified as significant were further analyzed by two methods, principal component analysis and boxplot analysis, to determine the distributions of normalized sequence read counts (reads per kilobase per million mapped reads) for each individual DNase-seq library. Three outlier DNase-seq samples were thus identified: two from livers designated STAT5-high activity based on EMSA (samples G74A_M1 and G74A_M2), which gave DNase-seq read count distributions (top 200 and top 600 diffReps-identified differential sites, ordered by decreasing fold-change in chromatin opening) more similar to STAT5-low-activity livers; and one from a STAT5-low EMSA activity liver (sample G92_M5), which gave a DNase-seq read count distribution more similar to STAT5-high-activity livers. The three outlier liver DNase-seq libraries were excluded from all downstream analysis. Next, we implemented diffReps analysis using the same cutoffs described above to compare the DNase-seq activity profiles of the remaining EMSA-identified STAT5-high-activity (n = 8) and STAT5-low-activity (n = 10) DNase-seq samples. The resultant set of diffReps differential sites

was overlapped with a MACS2 DHS peak union list, which was obtained by merging the MACS2 DHS peak calls from all 18 male mouse liver DNase-seq samples using the BEDTools *Merge* command. This overlap analysis yielded a final set of 70,767 MACS2 DHS peak union sites (*Supplementary file 5B*), of which 2832 MACS2-identified DHS were more open (more accessible state) and 123 DHS were in a more closed state based on their overlap with the diffReps-identified STAT5-high vs. STAT5-low livers differential regions (*Figure 2E*). The other 67,812 DHS regions were designated static DHS as they did not overlap a diffReps-identified STAT5-high vs. STAT5-low differential region.

A comparison of these 70,767 MACS2-identified DHS peak union sites with our standard reference set of 70,211 mouse liver DHS identified n = 2373 reference set DHS that overlapped a STAT5-high differential site (*Figure 2F*), n = 98 reference set DHS that overlapped a STAT5-low differential site, and n = 45,754 liver DHS that overlapped a static DHS (*Supplementary file 1A*, columns H and I). The remaining 21,986 standard reference set DHS did not overlap the 70,767 DHS peak union list and were also labeled static DHS. We applied the designation of sex bias determined previously (*Ling et al., 2010*), namely, male-biased, female-biased, or sex-independent, to each DHS that overlapped the reference set 70,211 liver DHS (*Supplementary file 1A*, column F). The standard reference set liver DHS were further designated dynamic DHS if they overlapped a STAT5-high > STAT5-low differential DHS (i.e., a GH/STAT5 pulse-opened DHS). Liver DHS not identified as dynamic (including the 98 STAT5-low > STAT5-high DHS peak union sites that overlap a standard reference set DHS) were designated static with respect to chromatin accessibility changes induced by endogenous GH-induced STAT5 pulses. Sex-specific DHS were thus designated dynamic male-biased DHS, static male-biased DHS, dynamic female-biased DHS, or static female-biased DHS with respect to GH/STAT5-induced chromatin opening (*Supplementary file 1A*, column I). Similarly, sex-independent DHS were designated dynamic sex-independent or static sex-independent (*Figure 2F*).

## DNase-I cut site aggregate plots

Normalized DNase-I cut site aggregate plots were generated using an input DNase-seq dataset and the set of input genomic regions (DHS sequences) to be analyzed by sequence read counting. First, FASTQ files from DNase-seq biological replicates were concatenated to obtain a single combined replicates file for each treatment group. For example, for male liver, we generated a single combined STAT5-high DNase-seq FASTQ file by merging the n = 8 biological replicate STAT5-high liver FASTQ files, and separately, we generated a single combined STAT5-low sample by merging the n = 10 biological replicate STAT5-low DNase-seq FASTQ files (*Supplementary file 4*). Combined replicate FASTQ files were generated for intact female, hypox female, hypox male, and each of the hypox male + GH treatment time point DNase-seq samples in the same manner. Second, for each combined replicate file, a BED file comprised of positive and negative strand reads was processed to identify each DNase-I cut site, which corresponds to the 5'-end of each sequence read. Third, the set of input genomic regions was processed to generate a list of 2 kb regions centered at the midpoint of each DHS. The BEDTools *Coverage* command using the (-d) option was then used to count the number of DNase-I cut sites at each nt position across each 2 kb midpoint-centered region, thus producing a read count matrix composed of 2000 read counts for each input genomic region. Fourth, a custom R script was used to load the read count matrix, calculate the sum of read counts at each nt position, normalize the raw read counts by the number of reads in the subset of 70,211 standard reference DHS to be analyzed, normalize by the number of input genomic regions, and then generate a plot of the DNase-I signal across the input genomic regions. The resultant normalized DNase-I cutting profiles were then smoothed using a LOWESS smoother as implemented in R (package: gplots v3.0.1.1). An offset was applied to the profile by subtracting an average read count (calculated from the first 200 nt positions) from the normalized read count intensity to standardize the baseline of each profile. Profiles for other DNase-seq datasets were processed in the same manner and were plotted on the same set of axes to enable direct comparisons across all such plots (e.g., *Figures 2, 3 and 5*).

## Impact of hypophysectomy and GH replacement on chromatin opening and closing

MACS2 was used to discover DHS peaks in DNase-seq samples prepared from intact male and intact female mouse liver, hypox male and hypox female mouse liver, and from livers of hypox male mice given a single replacement dose of GH and euthanized either 30, 90, or 240 min later (*Supplementary*

*file 4*). The effects of hypophysectomy on chromatin accessibility were determined by comparing hypox liver DNase-seq samples to the corresponding samples prepared from intact liver (control) samples in male liver and separately in female liver. The effects of a single injection of GH were determined by comparing DNase-seq samples from livers of hypox male mice treated with GH to the untreated hypox male liver controls at each time point. For each comparison, a DHS peak union list was generated by merging the MACS2 DHS peak calls from all individual biological replicates for the corresponding treatment group. For example, a single peak union list for the hypox male compared to intact male liver samples was generated by merging all the MACS2 peaks from each of the n = 8 male mouse liver DNase-seq samples (n = 6 intact individual male control liver libraries and n = 2 hypox male liver libraries, each prepared from a pool of n = 3 individual livers; *Supplementary file 4*). Genomic regions that showed significantly differential DNase-seq signal between hypox and intact control livers, or between hypox male + GH and hypox male livers were discovered separately for each comparison using diffReps using the nucleosome option (200 bp window). Significance was based on |fold-change| > 2 and FDR < 0.05 (Benjamini–Hochberg adjusted p-value) for diffReps-normalized signal intensity values. The diffReps-identified differential sites were then filtered by their overlap with the peak union list for all eight samples to obtain the sets of differential DHS (e.g., 2142 DHS that open and 4856 DHS that close in male liver following hypophysectomy; *Supplementary file 5A*). MACS2 peak union DHS that did not overlap a diffReps region were annotated as static DHS peaks (50,055 sites). Concatenation of the diffReps-identified differential DHS with this set of static DHS yielded the full set of 57,053 DHS peak union sites for this dataset. Each of the 70,211 standard reference set liver DHS (see above) was then labeled based on whether it overlapped a diffReps-identified differential DHS that opened following hypophysectomy, a diffReps differential DHS that closed following hypophysectomy, a static DHS, or 'none,' for those reference set DHS that did not overlap any of the set of 57,053 hypophysectomy study DHS peak union sites. Corresponding analyses were performed for the intact and hypox female liver samples, and for the hypox male + GH time-course comparisons mentioned above. *Supplementary file 5A* summarizes the numbers of hypophysectomy and hypophysectomy + GH responsive differential DHS for each of these comparisons, and the subsets that overlap the reference set of 70,211 liver DHS; full peak lists are provided in *Supplementary file 5C–G*.

## Sex-biased genes and hypophysectomy response classification

RNA-seq data from intact male and intact female mouse liver was previously used to identify sex-biased genes based on a gene list comprised of 24,197 RefSeq genes (*Connerney et al., 2017*). This list was expanded to include both RefSeq and multi-exonic sex-specific lncRNA genes as follows. Sex-specific RefSeq genes were identified from polyA-selected total liver RNA-seq samples from intact male and intact female liver using the 'genebody' method of sequence read counting (*Connerney et al., 2017*) at a threshold of |fold-change| > 1.5, edgeR-determined adjusted p-value<0.05, and FPKM > 0.25 for the sex with a greater signal intensity, which yielded 387 male-specific and 517 female-specific RefSeq genes. Sex-specific multi-exonic lncRNA genes were identified from polyA-selected liver nuclear RNA-seq samples in male and female mouse liver using the 'ExonCollapsed' read counting method (*Melia and Waxman, 2019*) at a threshold of |fold-change| > 2, adjusted p-value<.05 and FPKM > 0.25 for the sex with a higher signal intensity, which yielded 121 male-specific and 102 female-specific multi-exonic lncRNA genes, for a total of 508 male-specific and 619 female-specific genes (*Supplementary file 2*). A stringent set of sex-independent genes was defined by FPKM > 1 in both male and female liver, |fold-change| < 1.2, and adjusted p-value>0.1, which yielded a total of 7253 stringently sex-independent genes, including lncRNA genes (*Supplementary file 2*, column N). Sex-biased genes responsive to hypophysectomy were defined by |fold-change| > 2 and edgeR-determined adjusted p-value<0.05 for hypox mouse liver vs. intact mouse liver, determined separately for male liver and female liver. Using these thresholds, RNA-seq data from intact and hypox male and female liver samples were used to assign sex-biased genes into classes I and II and their corresponding subclasses (IA, IB, IC, IIA, and IIB) (*Wauthier et al., 2010*; *Connerney et al., 2017*). Class I sex-biased genes are those sex-biased genes that are downregulated by hypophysectomy in the sex where they show higher expression in intact mice. Class II sex-biased genes are those that are upregulated by hypophysectomy in the sex where they show lower expression in intact mice (see model in *Figure 4—figure supplement 1A*). Subclasses A, B, and C indicate the response to hypophysectomy in the dominant

sex (class II genes) or in the opposite sex (class I genes), as defined in Table 3 of *Wauthier et al., 2010*. The number of sex-biased genes in each hypophysectomy response class is summarized in *Supplementary file 2*.

## STAT5 binding and motif analysis

ChIP-seq analysis of STAT5 binding in male and female mouse liver identified 15,094 merged peaks comprised of male-enriched and female-enriched, and male-female common STAT5 binding sites (*Zhang et al., 2012*). A small subset of these STAT5 binding sites, 75 peaks, overlapped ENCODE blacklisted regions and was excluded from downstream analysis. BEDTools overlap analysis of these STAT5 binding sites allowed us to designate each of the 70,211 reference DHS as STAT5-bound if STAT5 binding was observed in the ChIP-seq dataset, or as 'not bound.' *Supplementary file 1A* (columns J–N) provides full details, including the STAT5 binding site, the sex specificity of STAT5 binding (male-enriched, female-enriched, or common to both sexes), the normalized ChIP-seq read counts for STAT5 binding in STAT5-high-activity male livers (MH) and STAT5-high-activity female livers (FH), and the average of these two sets of read counts. The STAT5B motif M00459 from the TRANSFAC motif database (release 2011.1) (*Matys et al., 2006*) was used to determine the frequency of STAT5 motif occurrence in each set of DHS sequences. Motifs found in DHS sequences were identified using FIMO (v4.12.0) (*Grant et al., 2011*) using the option (--thres 0.0005) to improve detection of short length motifs. The number of STAT5B motif occurrences in each of the 70,211 reference set DHS sequences is shown in *Supplementary file 1A*, column O.

## Enrichment analysis

For all enrichment calculations described below, the significance of the enrichment was determined by a Benjamini–Hochberg adjusted Fisher's exact test p-value as implemented in R. Enrichments with adjusted p-value<1E-03 were considered statistically significant. Enrichments of sex-biased DHS for being enhancer, promoter, or insulator regions (e.g., male-biased DHS for being enhancers compared to the background set of sex-independent DHS) were calculated as follows: enrichment score = (ratio A)/(ratio B), where ratio A = the number of sex-biased DHS that are enhancers, divided by the number of sex-biased DHS that are not enhancers; and ratio B = the number of sex-independent DHS that are enhancers, divided by the number of sex-independent DHS that are not enhancers. For example, 2551 male-biased DHS were classified as enhancers, and 178 other male-biased DHS were not enhancers (2551/178 = 14.3), whereas 43,591 sex-independent DHS were classified as enhancers, and 22,525 sex-independent DHS were not enhancers (43,591/22,525 = 1.93), which gives an enrichment score A/B = 14.3/1.93 = 7.41. The set of 66,116 sex-independent DHS was used as the background for these enrichment calculations.

Enrichments of sex-biased DHS subsets (enhancer, insulator, or promoter) for mapping to sex-biased genes were calculated (e.g., male-biased enhancer DHS mapping to male-specific genes) as follows (*Supplementary file 1B*): enrichment score = (ratio A)/(ratio B), where ratio A = the number of male-biased enhancer DHS that map to male-specific genes, divided by the number of male-biased enhancer DHS that map to sex-independent genes; and ratio B = the number sex-independent enhancer DHS that map to male-specific genes, divided by the number of sex-independent enhancer DHS that map to sex-independent genes. For example, among the male-biased DHS classified as enhancers, 404 male-biased enhancer DHS map to male-specific genes, and 525 male-biased enhancer DHS map to sex-independent genes (404/525 = 0.77), whereas 1495 sex-independent enhancer DHS map to male-specific genes, and 15,457 sex-independent enhancer DHS map to sex-independent genes (1495/15,457 = 0.097), which gives an enrichment score A/B = 0.77/0.097 = 8.0. The set of 66,116 sex-independent DHS was used as the background for these enrichment calculations.

Enrichments of hypophysectomy-responsive DHS for mapping to class I or II sex-biased genes (e.g., DHS that close in response to hypophysectomy in male liver and that map to male class I genes) were calculated for male and female liver as follows: enrichment score = ratio A/ratio B, where ratio A = the number of DHS that open (or that close) following hypophysectomy and that map to class I (or to class II) sex-biased genes, divided by the number of DHS that open (or that close) following hypophysectomy and that map to sex-independent genes; and ratio B = the number of hypophysectomy-unresponsive DHS (static DHS) that map to class I or II sex-biased genes, divided by the number of static DHS that map to sex-independent genes. For example, in male liver, 217 DHS that close

following hypophysectomy map to a male class I sex-specific gene, and 1203 other DHS that close map to sex-independent genes (217/1203 = 0.1803), whereas 444 static DHS map to a male class I sex-specific gene, and 14,053 static DHS map to a sex-independent gene (444/14,053 = 0.0316), which gives an enrichment score A/B = 0.1803/0.0316 = 5.7. The static DHS used for these enrichment calculations correspond to the set of 35,562 static DHS in male liver and 30,394 static DHS in female liver.

Enrichments for overlap with genomic regions containing biologically relevant sets of transcription factor binding sites, chromatin marks, and combinations of epigenetic features (chromatin states) were calculated for each of the following four sets of DHS: (1) dynamic male-biased DHS (834 sites), (2) static male-biased DHS (1895 sites), (3) dynamic sex-independent DHS (1532 sites), and (4) static female-biased DHS (1359 sites) relative to a background set of static sex-independent DHS (64,584 sites) (see *Figure 2F*). Briefly, these sets of DHS were identified based on their sex specificity (*Ling et al., 2010*) and were classified as 'dynamic' if they overlapped a STAT5-high vs. STAT5-low differential DHS, otherwise they were classified as 'static.' Characterization of the 70,211 reference set DHS members as static or dynamic and their sex-specificity designations are listed in *Supplementary file 1A*. Biologically relevant regions annotated include sex-biased transcription factor binding sites for STAT5 and BCL6 (*Zhang et al., 2012*), CUX2 (*Conforto et al., 2012*) and FOXA1/FOXA2 (*Li et al., 2012*). We also examined DHS enrichment for sex-biased chromatin marks and chromatin states defined for male and female liver (*Sugathan and Waxman, 2013*). FOXA1 and FOXA2 ChIP-seq data (*Li et al., 2012*) was processed and reanalyzed with MAnorm (*Shao et al., 2012*) in analyses performed by Gracia Bonilla of this laboratory, which identified sex-biased FOXA1 and sex-biased FOXA2 binding sites in mouse liver. Data from the MAnorm analysis of FOXA1 and FOXA2 ChIP-seq peaks (*Supplementary file 8*) was filtered by log2(fold-change) value (i.e., M-value) to define male-biased and female-biased binding sites. Genomic coordinates for each of the sex-biased transcription factor binding sites, chromatin marks, and chromatin states are provided (*Supplementary files 3 and 7*). Enrichments for the overlap of DHS with various biologically relevant regions were calculated (e.g., dynamic male-biased DHS for containing STAT5-high [male] binding sites compared to the background set of static sex-independent DHS) as follows: enrichment score = ratio A/ratio B, where ratio A = the percentage of dynamic male-biased DHS with a STAT5 binding site; and ratio B = the percentage of static sex-independent DHS with the same type of binding site. For example, 235 dynamic male-biased DHS contain a STAT5-high (male) binding site (235/834 = 0.2818), whereas 311 static sex-independent DHS contain such a binding site (311/64,584 = 0.0048), which gives an enrichment score A/B = 0.2818/0.0048 = 58.7. The set of 64,584 static sex-independent DHS (*Figure 2F*) was used as the background for these enrichment calculations.

## Chromatin state map analysis

Chromatin state maps (14 state model), developed for male mouse liver, and separately for female mouse liver, are based on epigenetic data from a panel of six histone marks and DHS data in each sex (*Sugathan and Waxman, 2013*). These maps were used to identify sex differences in chromatin state and chromatin structure at each DHS region and their relationships to sex-biased gene expression as follows. BEDTools was used to assign one of the 14 chromatin states to each of the 70,211 reference DHS based on the overlap of the DHS with the male liver chromatin state map, and separately, based on its overlap with the female liver chromatin state map (*Supplementary file 1A*, columns AN–AQ). DHS whose genomic coordinates span two or more different chromatin states were assigned to the state with the largest number of overlapping base pairs, or in case of equal numbers of base pairs, the chromatin state with the smaller genomic coordinates. Enrichments of sets of static and dynamic male-biased DHS for being in one of the 14 chromatin states in male liver (e.g., dynamic male-biased DHS for being in chromatin state E6, which is an enhancer-like state) were calculated as follows: enrichment score = (ratio A)/(ratio B), where ratio A = the number of male-biased DHS that are in a particular chromatin state, divided by the number of male-biased DHS not in that chromatin state; and ratio B = the number of static sex-independent DHS in that chromatin state, divided by the number of static sex-independent DHS not in that chromatin state. For example, 604 dynamic male-biased DHS are classified as chromatin state E6, and 230 other dynamic male-biased DHS are not classified as chromatin state E6 (604/230 = 2.626), whereas 24,082 static sex-independent DHS are classified as chromatin state E6, and 40,502 static sex-independent DHS are not classified as chromatin state

E6 (24,082/40,502 = 0.5946), which gives an enrichment score A/B = 2.626/0.5946 = 4.4. The set of 64,584 static sex-independent DHS was used as the background set for these enrichment calculations.

## DHS peak normalization

DNase-seq data to be visualized in the UCSC Genome browser (https://genome.ucsc.edu/) was normalized using the number of sequence reads in each DHS peak region per million mapped sequence reads (reads-in-peaks-per-million [RiPPM]) as a scaling factor (*Lodato et al., 2018*). Normalization was carried out using a comprehensive list of DHS peak regions merged across each dataset (peak union list), obtained by concatenating FASTQ files for biological replicates of each control and treatment group, as described above. DHS peak regions identified in both individual liver samples and by analysis of the combined samples were concatenated into a single list, and then the BEDTools *Merge* command was used to combine overlapping features to generate a single list of non-overlapping DHS peaks. The fraction of reads in peaks for each sample was then calculated to obtain a scaling factor. Raw sequence read counts were divided by this per-million scaling factor to obtain RiPPM normalized read counts.

## Acknowledgements

This work was supported in part by NIH grant DK121998 (to DJW).

## Additional information

### Funding

| Funder | Grant reference number | Author |
|---|---|---|
| National Institutes of Health | NIDDK R01DK121998 | David J Waxman |

The funders had no role in study design, data collection and interpretation, or the decision to submit the work for publication.

### Author contributions

Andy Rampersaud, Data curation, Software, Formal analysis, Investigation, Visualization, Methodology, Data processing, primary computational analysis, preparation of analysis scripts, and data visualization were carried out by AR. All other data analysis and preparation of datasets and figures for publication were carried out by AR and DJW. AR and DJW jointly prepared a preliminary draft of the manuscript; Jeannette Connerney, All animal studies and wet lab analyses including STAT5 EMSA gels, isolation of liver nuclei, DNase-seq analysis and sequencing library preparation were performed by JC; David J Waxman, Data curation, Formal analysis, Funding acquisition, Investigation, Visualization, Methodology, Project administration, DJW conceived of the study with input from AR. Secondary data analysis and preparation of datasets and figures for publication were carried out by AR and DJW. AR and DJW jointly prepared a preliminary draft of the manuscript. The final manuscript was written by DJW and was reviewed and approved by all the authors. DJW supervised the overall project and revised edited the final manuscript for publication

### Author ORCIDs

David J Waxman ORCID https://orcid.org/0000-0001-7982-9206

### Ethics

All mouse work was carried out in accordance with ARRIVE Essential guidelines 2.0 for study design, sample size, randomization, experimental animals and procedures, and statistical methods, and with approval of the Boston University Institutional Animal Care and Use Committee (protocol # 16-003).

Reviewer #1 (Public Review): https://doi.org/10.7554/eLife.91367.3.sa1
Reviewer #2 (Public Review): https://doi.org/10.7554/eLife.91367.3.sa2
Author Response https://doi.org/10.7554/eLife.91367.3.sa3

# Additional files

## Supplementary files

• Supplementary file 1. Liver DHS. (A) Standard reference set of 70,211 liver DHS regions and associated datasets. (B) Enrichments of sex-biased DHS for corresponding sex-biased vs. sex-opposite gene targets (sections 1–3) and enrichments for hypox-responsive class I and II sex-biased genes (section 4). (C) Summary of sex-biased gene targets of sex-biased DHS, from sheet A. Columns A and B indicate that multiple sex-biased DHS map to many sex-biased genes. The overall set of sex-biased DHS mapped to a total of 192 male-biased genes and 174 female-biased genes, corresponding to 32.5% of all sex-biased genes considered. (D) Calculation of chromatin state distributions shown in *Figure 1D*, based on data in sheet A, columns AN–AQ. (E) DHS closing and opening following continuous GH infusion in male mice for 7 d, based on data in sheet A, column BC.

• Supplementary file 2. Sex-biased genes. Shown are mouse liver gene expression for RefSeq and multi-exonic lncRNA genes, their sex-bias classification, and hypophysectomy response classes. Data shown are normalized differential expression ratios, calculated as intact-male/intact-female, corresponding fold-change (FC) values, normalized read counts (FPKM; fragments per kilobase of region of interest per million mapped reads) for male and female liver, and FDR (adjusted p-values) for the comparison.

• Supplementary file 3. Chromatin states in male (A) and in female (B) mouse liver. Chromatin state maps (14 state model) were previously developed for male mouse liver, and separately for female mouse liver, using a panel of six histone marks and DHS data, and used to identify sex differences in chromatin state and chromatin structure and their relationships to sex-biased gene expression (*Sugathan and Waxman, 2013*). BEDTools was used to determine the overlap between the 14 chromatin states identified in male liver and the reference set of 70,211 DHS used in this study. The genomic position of the chromatin state assignment is listed in columns A–C, corresponding name (column D), a unique ID (CS Number, column E), and its overlapping DHS region (columns F–I). The number of overlapping base pairs is provided in column J; values of zero indicate lack of DHS overlap for that chromatin state region.

• Supplementary file 4. Summary of DNase-seq analysis. Shown are the total and mapped read counts for the DNase-seq samples prepared and analyzed in this study, the number of DHS peaks discovered in each sample by MACS2, and the fraction of sequence reads found in the sample's respective peak list. The data for the pituitary-intact male liver (STAT5-high and STAT5-low) samples and the GH time-course mouse liver (hypophysectomized [hypox] and GH treated) samples are shown separately.

• Supplementary file 5. Hypophysectomy-responsive DHS. (A) DHS peak summary and enrichment calculations summary for the sets of DNase-seq peaks discovered for the pituitary-intact male liver (STAT5-high and STAT5-low) samples and for the GH time-course mouse liver (hypophysectomized [hypox] and GH treated) samples. Each sheet (B–G) contains the peak union list generated from the respective data set and indicates a single overlapping DHS for each peak union site. Also shown is a summary of the enrichments shown in sheet I and presented in *Figure 4D*. (B–G) Shown for each indicated DHS set is the DHS overlap with the reference set of 70,211 DHS. Merged list of 70,767 DHS regions was generated based on the MACS2 peaks called for each DNase-seq replicate sample and assigned a unique name (DHS Peak Number, column E). The genomic position of the DHS region is listed in columns A–C, its DHS response in column D, and its overlap with the reference set of 70,211 DHS is shown in columns (F–I). The number of overlapping base pairs is provided in column J; values of zero indicate lack of overlap between DHS regions. (H) Responsiveness of sex-biased DHS classes to hypox in male and in female mouse liver. DHS counts are based on sex-biased DHS classifications in *Supplementary file 1A*, column I, and hypophysectomy response data summarized in columns AI–AJ of that sheet. No DHS overlap indicates that the indicated male or female hypox dataset does not contain the indicated number of DHS from the standard reference set of 70,211 liver DHS listed in *Supplementary file 1A*. For example, 19 of 834 dynamic male-biased DHS and 454 of 1895 static male-biased DHS are absent from the male hypox DHS dataset used for these analyses. Those DHS were excluded when calculating the percentage values shown in columns C, E, G, etc. (I) Hypophysectomy enrichments of DHS regions.

• Supplementary file 6. Summary of DNase-seq aggregate plots. Shown is the maximum value for each DNase-I cut site aggregate plot for each of the DNase-seq samples indicated, organized by manuscript figure number. Briefly, normalized DNase-I cut site aggregate plots were generated using input DNase-seq datasets (columns C–H) and the set of input genomic regions (DHS

sequences) (columns A–B) used for sequence read counting. A single replicate combined sample was generated for each of the following groups: male STAT5-high, male STAT5-low, female control, hypophysectomized male (MHx) and female (FHx), and MHx treated with GH (MHx + GH) at the following time points: 30, 90, and 240 min. These replicate combined samples were then used to determine the number of DNase-I cuts at each nucleotide position of the 2 kb midpoint-centered regions of the DHS region. The maximum value of the smoothed cumulative DNase-I cutting profile in each of the figures is shown below. All the numbers shown in columns C–H are directly comparable to each other: the data for aggregate profiles and also the RiPPM read counts shown in *Supplementary file 1* are normalized by the number of DNase-seq reads in the reference set of 70, 211 DHS (i.e., normalization by reads in peaks).

• Supplementary file 7. Transcription factor and histone marks enrichments. (A) Summary of genomic regions with transcription factor binding or with histone marks used for DHS enrichment calculations. (B) Summary of DHS enrichment calculations for the biologically relevant regions defined in sheet A. (C) Full listing of DHS enrichment calculations shown in sheet B. Shown is the calculated enrichment score (ES), the number and percent of overlapping DHS, the number and percent of overlapping background DHS, and Fisher's exact test p-value for the overlap between the DHS and each set of biologically relevant regions. (D–F) Coordinates for each of the sex-biased transcription factor binding sites (D) and chromatin mark regions in male liver (E) and in female liver (F), as defined in sheet A.

• Supplementary file 8. MAnorm comparative analysis of FoxA1 (A) and FoxA2 (B) ChIP-seq peaks between male and female mouse liver. Raw data from *Li et al., 2012* was processed and reanalyzed with MAnorm to identify sex-specific FoxA1 and FoxA2 binding sites in mouse liver. FoxA ChIP-seq samples in male and female liver were defined as samples 1 and 2, respectively. Shown is the MAnorm output which lists the peak coordinates, raw reads, M and A values, and MA-norm p-value for the set of common peaks in each sample and for the peaks unique to each sample. M-values are defined as the log2 fold change and A-values are defined as the average signal strength of the of normalized read densities under comparison.

• Supplementary file 9. Gene to DHS associations determined by GREAT analysis. (A–E) Gene-centric presentation of DHS regions that map to each gene; (F–I) DHS-centric presentation of genes that map to each DHS region.

• Supplementary file 10. GREAT analysis of four DHS sets of interest. (A–D) Full output from GREAT analysis of each of the four indicated DHS sets using a whole-genome background. Analysis was performed using GREAT version 4.0.4. (E) Mouse mm9 coordinates of each DHS set input to GREAT for the analyses shown in A–D.

• MDAR checklist

## Data availability

Raw and processed data for newly generated DNA sequence data are available at https://www.ncbi.nlm.nih.gov/gds under accession numbers GSE131848 and GSE131852 (SuperSeries GSE131853). Supplementary File 4 includes full details about all sequencing samples. Full datasets and associated data are provided in Supplemental Excel worksheets comprising Supplementary Files 1-10.

The following datasets were generated:

| Author(s) | Year | Dataset title | Dataset URL | Database and Identifier |
|---|---|---|---|---|
| Rampersaud A, Connerney J, Waxman DJ | 2023 | Changes in chromatin accessibility in male mouse liver induced by naturally occurring endogenous pulses of plasma growth hormone (GH)-activated STAT5 | https://www.ncbi.nlm.nih.gov/geo/query/acc.cgi?acc=GSE131848 | NCBI Gene Expression Omnibus, GSE131848 |
| Rampersaud A, Connerney J, Waxman DJ | 2023 | Changes in chromatin accessibility due to hypophysectomy (hypox) and a single exogenous pulse of GH/STAT5 in mouse liver | https://www.ncbi.nlm.nih.gov/geo/query/acc.cgi?acc=GSE131852 | NCBI Gene Expression Omnibus, GSE131852 |

The following previously published datasets were used:

| Author(s) | Year | Dataset title | Dataset URL | Database and Identifier |
|---|---|---|---|---|
| Li Z | 2012 | Foxa1 and Foxa2 are essential for gender dimorphism in liver cancer | https://www.ebi.ac.uk/biostudies/arrayexpress/studies/E-MTAB-805 | ArrayExpress, E-MTAB-805 |
| Ling G | 2010 | Unbiased, Genome-wide in vivo Mapping of Transcriptional Regulatory Elements Reveals Sex Differences in Chromatin Structure Associated with Sex-specific Liver Gene Expression | https://www.ncbi.nlm.nih.gov/geo/query/acc.cgi?acc=GSE21777 | NCBI Gene Expression Omnibus, GSE21777 |
| Zhang Y | 2012 | Dynamic, sex-differential STAT5 and BCL6 binding to sex-biased, growth hormone-regulated genes in adult mouse liver | https://www.ncbi.nlm.nih.gov/geo/query/acc.cgi?acc=GSE31578 | NCBI Gene Expression Omnibus, GSE31578 |
| Conforto TL | 2013 | Identification of Cux2 binding sites in female and male mouse liver | https://www.ncbi.nlm.nih.gov/geo/query/acc.cgi?acc=GSE35985 | NCBI Gene Expression Omnibus, GSE35985 |
| Sugathan A | 2013 | Genome-wide maps of histone modifications in male and female mouse liver | https://www.ncbi.nlm.nih.gov/geo/query/acc.cgi?acc=GSE44571 | NCBI Gene Expression Omnibus, GSE44571 |

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
