## [Editor Report · eLife assessment]

This **important** study offers new and **convincing** support for the idea that about a third of mouse liver DNAse-I hypersensitivity sites (DHS) showing male-biased chromatin opening are sex-biased because of the male-specific cyclic action of growth hormone pulses to alter chromatin accessibility compared to the relative ineffectiveness of the more static pattern of growth hormone secretion in females. Supporting evidence is found in the impact of hypophysectomy and growth hormone treatment on chromatin accessibility, and the binding of specific transcription factors and epigenetic marks at STAT5-sensitive sites. This work uncovers mechanisms underlying sex differences in liver function and will be of broad interest to endocrinologists and hepatologists.

---

## [Referee Report · Reviewer #1 (Public Review)]

Summary:

Sex differences in the liver gene expression and function have previously been proposed to be caused by sex differences in the pattern growth hormone (GH) secretion by the pituitary, which are established by the effects of testicular hormones that act on the hypothalamus perinatally to masculinize control of pituitary GH secretion beginning at puberty and for the rest of the animal's life. The Waxman lab has previously implicated GH control of STAT5 as a critical event leading to a masculine pattern of gene expression. The present study separates male-biased regulatory sites associated with the male-biased genes into different classes based on their responsiveness to the cyclic male pattern of STAT5 activity, and investigates DNAse hypersensitivity sites (DHS) of different classes showing cyclic sex-bias or not. It further reports on the binding of transcription factors to STAT5-sensitive DHS, and involvement of specific histone marks at these sites. The study argues that STAT5 is the proximate factor regulating chromatin accessibility in about 1/3 of male-biased DHS that are sexually differentiated by GH secretion. The authors propose the pulsatile GH secretion as a novel proximate mechanism of regulating chromatin accessibility to cause sex differences.

Strengths:

The study offers new insight into the effects of hypophysectomy and injection of GH on different classes of sex-biased genes in mouse liver. The results support the general conclusion of the authors. Cyclic secretion of other hormones (for example, estrous secretion of estrogens and progesterone) are well known to cause sex differences in multiple organs in rodents, and it will be interesting to assess if these cyclic secretions induce similar changes in chromatin accessibility causing female tissue gene expression to differ from that of males.

Weaknesses:

The authors argue for two major mechanisms controlling sexual bias in liver gene expression, and analyze in depth one of these mechanisms. The focus is on the group of DHS (about 1/3 of all male-biased DHS) in which the sex bias is controlled by cyclic secretion of growth hormone (GH) in males, compared to static and low growth hormone in adult females. The sex difference in pituitary secretion of GH is induced by permanent effects of androgens acting on the hypothalamus perinatally. The manuscript study would be improved by further discussion of the mechanistic relationship between this class of sex-biased DHS and the other 2/3 of liver DHS that also show male-biased accessibility but whose chromatin does not respond directly to GH-stimulated STAT5. Previous studies, including those in the Waxman lab (PMIDs: 26959237, 18974276, 35396276) suggest castration of males or gonadectomy of both sexes eliminates most sex differences in mRNA expression in mouse liver, and/or that androgens such as DHT or testosterone administered in adulthood potentially reverses the effects of gonadectomy and/or masculinizes liver gene expression. It is not clear from the present discussion whether the GH/STAT5 cyclic effects to masculinize chromatin status require the presence of androgens in adulthood to masculinize pituitary GH secretion. Are there analyses of the present (or past) data that might provide evidence about a dual role for GH and androgen acting on the same genes? For example, are sex-biased DHS bound by androgen-dependent factors or show other signs of androgen sensitivity? Are histone marks associated with DHS regulated by androgens? Moreover, it would help if the authors indicate whether they believe that the "constitutive" static sex differences in the larger 2/3 set of male-biased DHS are the result of "constitutive" (but variable) action of testicular androgens in adulthood. Although the present study is nicely focused on the GH pulse-sensitive DHS, is there mechanistic overlap in sex-biasing mechanisms with the larger static class of sex-biased liver DHS?

---

## [Referee Report · Reviewer #2 (Public Review)]

Summary:

The present work addresses the mechanisms linking the sex-dependent temporal GH secretion patterns to the robust sex differences in chromatin accessibility and transcription factor binding that ultimately regulate sexually dimorphic liver gene expression. Using DNAseq analysis genomic sites hypersensitive to cleavage by DNase I, DNase hypersensitive sites [DHS] were studied in hepatocytes from male and female mice. DHS in the genome correspond to accessible chromatin regions and encompass key regulatory elements, including enhancers, promoters, insulators, and silencers, often flanked by specific histone modifications, and all of these players were described in different settings of GH action. Importantly, the dynamics of sex-dependent and independent chromatin accessibility linked to STAT5 binding were evaluated. For that purpose, hepatic samples from mice were divided into STAT high and STAT low binding by EMSA screening. With this information changes in DHS related to STAT binding were calculated in both sexes, giving an approximation of chromatin opening in response to STAT5, or alternatively to hypophsectomy, or a single GH pulse. More the 800 male-biased DHS (from a total of more than 70000 DHS) regions were identified in the STAT5 high groups, implying that the binding of a plasma GH pulse activates STAT5, and evokes a dynamic cycle of male liver chromatin opening and closing at sites that comprised 31% of all male-biased DHS. This proves that the pulsatility of plasma GH stimulation confers significant male bias in chromatin accessibility, and STAT5 binding at a fraction of the genomic sites linked to sex-biased liver gene expression and liver disease. As a proof of concept, authors show that a single physiological replacement dose or pulse of GH given to hypophysectomized mice recapitulate, within 30 min, the pulsatile re-opening of chromatin seen in pituitary-intact male mouse liver.

In another male-biased DHS set (69% of male-biased DHS), chromatin accessibility was static, that is unchanged across the peaks and valleys of GH-induced liver STAT5 activity and mapped to a set of target genes and processes distinct though sometimes overlapping those of the dynamic male-biased DHS.

In view of these distinct dynamic and static DHS in males, authors evaluated key epigenetic features distinguishing the dynamic STAT5-driven mechanism of chromatin opening from that of static male-biased DHS, which are constitutively open in the male liver but closed in the female liver. The analysis of histone marks enriched at each class of sex-biased DHS indicated exquisite differences in the epigenetic mechanisms that mediate sex-specific gene repression in each sex. For example, H3K27me3 and H3K9me3, two widely used repressive histone marks, are used in a unique way in each sex to enforce sex differences in chromatin states at sex-biased DHS.

Finally, the work recapitulates and explains the classifications of sex dimorphic genes made in previous works. Sex-biased and pituitary hormone-dependent DHS act as regulatory elements with a positive enhancer potential, to induce or maintain gene expression in the intact liver by sustaining an open chromatin in the case of class I male-biased DHS and class I male-biased genes in the male liver. Contrariwise DHS may participate in the inhibition of gene expression by maintaining a closed chromatin state, as in the case of class II male-biased DHS and class II female-biased genes in male liver.

These results as a whole present a complex mechanism by which GH regulates the sexual dimorphism of liver genes in order to cope with the metabolic needs of each sex. In a complete story, the information on chromatin accessibility, histone modification, and transcription factor binding was integrated to elucidate the complex patterns of transcriptional regulation, which is sexually dimorphic in the liver.

Strengths:

The work presents a novel insight into the fundamental underlying epigenetic mechanisms of sex-biased gene regulation.

Results are supported by numerous Tables, and Supplementary Tables with the raw data, which present the advantage that they may be reanalyzed in the future to prove new hypotheses.

Weaknesses

It is a complicated work to analyze, even though the main messages are clearly conveyed.

---

## [Author Response]

The following is the authors’ response to the original reviews.

**Reviewer #1 (Public Review):**
1. The manuscript study would be improved by further discussion of the mechanistic relationship between this class of sex-biased DHS and the other 2/3 of liver DHS that also show male-biased accessibility but whose chromatin does not respond directly to GH-stimulated STAT5.

Response: We added a new paragraph to the Discussion (lines 608-618) discussing our novel finding that sex-biased H3K36me3 marks uniquely distinguish Static sex-biased DHS from Dynamic sex-biased DHS (see Fig. 6C) in light of a recent study in a different biological system showing that H3K36me3 marks comprise an important mechanism for maintaining cell type-specific identity by inhibiting the spread of H3K27me3 repressive marks at cell type-specific enhancers [Nat Cell Biol, 25 (2023) 1121-1134]. Further, we now discuss the potential mechanistic significance of this mark in insuring the sex-biased chromatin accessibility at Static sex-biased DHS:

“Finally, we discovered that sex-biased H3K36me3 marks are a unique distinguishing feature of static sex-biased DHS, with male-biased H3K36me3 marks being highly enriched at static male-biased DHS but not at dynamic male-biased DHS, and female-biased H3K36me3 marks highly enriched at static female-biased DHS (Fig. 6C). H3K36me3 marks are classically associated with the demarcation of actively transcribed genes [50] but are also used to maintain cell type identity by inhibiting the spread of H3K27me3 repressive marks at cell type-specific enhancers [35, 51]. The enrichment of H3K36me3 marks at static male-biased DHS described here could thus be an important mechanism to maintain sex-dependent hepatocyte identity by keeping static male-biased enhancers constitutively open and free of H3K27me3 repressive marks in male liver, and similarly for H3K36me3 marks enriched at static female-biased DHS in female liver. Further study is needed to elucidate the underlying mechanisms whereby these and the other sex-specific histone marks discussed above are deposited on chromatin in a sex-dependent and site-specific manner and the roles that GH plays in regulating these epigenetic events”.

1. Previous studies, including those in the Waxman lab (PMIDs: 26959237, 18974276, 35396276) suggest castration of males or gonadectomy of both sexes eliminates most sex differences in mRNA expression in mouse liver, and/or that androgens such as DHT or testosterone administered in adulthood potentially reverses the effects of gonadectomy and/or masculinizes liver gene expression. It is not clear from the present discussion whether the GH/STAT5 cyclic effects to masculinize chromatin status require the presence of androgens in adulthood to masculinize pituitary GH secretion. Are there analyses of the present (or past) data that might provide evidence about a dual role for GH and androgen acting on the same genes? For example, are sex-biased DHS bound by androgen-dependent factors or show other signs of androgen sensitivity? Are histone marks associated with DHS regulated by androgens? Moreover, it would help if the authors indicate whether they believe that the "constitutive" static sex differences in the larger 2/3 set of male-biased DHS are the result of "constitutive" (but variable) action of testicular androgens in adulthood. Although the present study is nicely focused on the GH pulse-sensitive DHS, is there mechanistic overlap in sex-biasing mechanisms with the larger static class of sex-biased liver DHS?

Response: The Reviewer poses an intriguing set of question regarding the potential role of androgens in directly regulating, perhaps by working together with GH or GH-activated STAT5 at the level of chromatin, to co-regulate the set of Static male-biased DHS. We have now addressed these questions in full in a new Discussion paragraph, entitled, “Pituitary GH secretory patterns vs. gonadal steroids as regulators of sex-biased liver chromatin accessibility and gene expression” (lines 640-661), as follows:

“While testosterone has a well-established role in programming hypothalamic control of pituitary GH secretory patterns [9-11], it is also possible that androgens and estrogens could regulate sex differences in hepatocytes directly at the epigenetic or transcriptional level. However, our findings support the proposal that plasma GH patterns, and not gonadal steroids, dominate epigenetic control of liver sex differences. First, the ability of a single exogenous plasma GH pulse to rapidly reopen dynamic male-biased DHS closed by hypophysectomy – in the face of ongoing ablation of pituitary stimulated gonadal steroid production and secretion – implicates GH signaling per se in the direct regulation of chromatin accessibility for this class of male-biased DHS. Second, GH regulates the sex bias of static male-biased DHS as well, as evidenced by their widespread closure in male liver following continuous GH infusion (Table S2E). It is important to note, however, that hepatocyte-specific knockout of androgen receptor (AR) does, in fact, dysregulate ~15% of sex-biased genes, albeit with a much lower effect size than global AR knockout [52] due to the systemic disruption of the somatotropic axis and circulating GH secretory profiles [53, 54]. Conceivably, AR could regulate these genes by a direct binding mechanism, acting either alone or in concert with GH-activated STAT5 to keep chromatin open constitutively at a subset of static male-biased DHS, of which 32% undergo at least partial closure in male liver following hypophysectomy (Fig. 4C). Estrogen receptor (ERa) likely plays only a minor role in regulating sex-biased liver DHS enhancers, given the lack of effect of hepatocyte-specific ERa knockout on sex-biased liver gene expression [22] and our finding that only 12% of static female-biased DHS close in female liver following hypophysectomy, which decreases circulating estradiol levels [55].”.

**Reviewer #2 (Public Review):**

The Reviewer did not raise any points of criticism.

**Reviewer #2 Recommendations:**
Line 121. "highly enriched for genes of the corresponding sex bias" is unclear. Does this mean that the genes near the DHS have the same bias in level of transcription as the bias in open chromatin? Please clarify.

Response: Text was changed to: “were highly enriched for mapping to genes showing the corresponding sex bias in the level transcription, but not for genes whose expression shows the opposite sex bias”.

Line 161. "STAT5 activity-dependent patterns" seems not to be supported by the data. The patterns correlate with STAT5 activity, but the authors can't conclude that they depend on STAT5 activity based on these data alone.

Response: Text was changed to: “patterns of DNase-released fragments that correlate with STAT5 activity”

Line 171. "identify genomic regions where chromatin dynamically opens or closes in male mouse liver in response to GH pulse activation of STAT5" This statement assumes a causal relationship between STAT5 and the status of differential sites. The data do not support this assumption of causality, because the data correlate STAT5 with status of the differential sites.

Response: Text was changed to: “identify genomic regions where chromatin dynamically opens or closes in male mouse liver in close association with GH pulse activation of STAT5”.

Line 176. The "binary pattern" in figure 2D seems not to be as binary as the authors suggest. The blue and red samples overlap in their distribution, and the lower green samples are intermediate between most of the blue and red samples. The "arbitrary" dotted line suggests the binary status, but this line is less convincing because it is arbitrary and drawn by eye; some samples don't obey the binary dichotomy.

Response: Text was changed to: “This pattern, where individual male mouse livers largely show either high or low DNase-seq read count distributions at the top differential genomic sites, was also seen…”.

Line 224 "independent" also implies causality.

Response: No changes were made.

Line 284. The effects of hypophysectomy on liver chromatin accessibility is attributed here to the loss of GH secretions. Hypophysectomy will also reduce testicular androgen secretion. To what extent can the results of Hypox be attributed to STAT5-dependent mechanisms as opposed to the loss of androgens?

Response: This question is now discussed in full in the new Discussion section, entitled, “Pituitary GH secretory patterns vs. gonadal steroids as regulators of sex-biased liver chromatin accessibility and gene expression” (lines 640-661), as noted above.

Line 505. "euthanized between plasma GH pulses". The authors are making an inference here because I do not think they measured GH levels. It would be more accurate to say that the time of euthanasia is inferred to be between GH pulses based on the measurement of STAT5 which is GH-dependent.

Response: Text was changed to: “a time inferred to be between plasma GH pulses”.

**Reviewer #3 Recommendations:**
In Figure 1A the differences between female-biased enhancers and sex-independent enhancers seem greater than those comparing female-biased insulators and sex-independent insulators, and yet only the latter are significant. Please could you clarify?

Response: Figure legend was corrected to indicate that Enhancers + Weak Enhancers were analyzed as a single group. Furthermore, the location of the Enhancer asterisks above the bars on the figure was adjusted to reflect this.

Line 257, I could not find Table S1B.

Response: Text in Figure legend was corrected to specify Table S7A as the source of this data.

Line 265 "BCL6 binding was also enriched at dynamic sex-independent DHS (Table S7B)." The p-value of this enrichment was particularly high. Could this have a biological correlation?

Response: We cannot rule out that possibility.

Line 277 "identified a Fox family factor as a close match for one of the top enriched motifs in the set of 278 static but not in the set of dynamic male-biased DHS", Maybe authors could add that this holds true for FOXI1 and not for FOXD1.

Response: Text was changed to specify FOXI1 as the factor.

Line 368, please clarify the affirmation because in Table 1A we do not see the data of dynamic and static male-biased DHS, but only male-biased, female-biased, and sex-independent DHS subsets.

Response: Text was corrected to read: “Our initial analyses revealed no major differences between dynamic and static male-biased DHS regarding the distribution of enhancer vs insulator vs promoter classifications (Fig. S7A) or their overall chromatin state distributions (Fig. S7B)”.

Figure 7A and 7B. It would visually help the reader if in E1, E2, etc. you could include the short definitions (as in Figure 1B: Inactive, Inactive, Low signal, etc.)

Response: We thank the reviewer for this suggestion, and have now added the X-axis labels suggested by the Reviewer.

Line 570 The sentence was difficult to read "similar to E6, but unlike E6," Maybe removing the comma after "unlike E6" would help.

Response: Text has been edited to avoid this cumbersome construct. It now reads: “…characterized by a high frequency of same activating chromatin marks as chromatin state E6, i.e., H3K27ac and H3K4me1 (E9) or H3K27ac alone (E10), but unlike E6 they are both deficient in…”.

Other changes include revisions to the Abstract to take into account the new discussion concerning the impact of sex-biased H3K36me3 marks along with related and other revisions to the Discussion, and a revision to the manuscript Title to better capture its main message.